# THE KINETICS OF REASONING: HOW CHAIN-OF-THOUGHT SHAPES LEARNING IN TRANSFORMERS?

## ABSTRACT

Chain-of-thought (CoT) supervision can substantially improve transformer performance, yet the mechanisms by which models learn to follow and benefit from CoT remain poorly understood. We investigate these learning dynamics through the lens of *grokking* by pretraining transformers on symbolic reasoning tasks with tunable algorithmic complexity and controllable data composition to study their generalization. Models were trained under two settings: (i) producing only final answers, and (ii) emitting explicit CoT traces before answering. Our results show that while CoT generally improves task performance, its benefits depend on task complexity. To quantify these effects, we model the accuracy of the logarithmic training steps with a three-parameter logistic curve, revealing how the learning speed and shape vary with task complexity, data distribution, and the presence of CoT supervision. We also uncover a transient *trace unfaithfulness* phase: early in training, models often produce correct answers while skipping or contradicting CoT steps, before later aligning their reasoning traces with answers. Empirically, we (1) demonstrate that CoT accelerates generalization but does not overcome tasks with higher algorithmic complexity, such as finding list intersections; (2) introduce a kinetic modeling framework for understanding transformer learning; (3) characterize trace faithfulness as a dynamic property that emerges over training; and (4) show CoT alters internal transformer computation mechanistically.

## 1 INTRODUCTION

Modern transformer language models can externalize part of their computation during inference by producing chain-of-thought (CoT) traces of natural-language or symbolic intermediate steps preceding a final answer (Wei et al., 2022b). This behavior can be elicited by prompting and further strengthened by supervised fine-tuning or reinforcement learning from human/model feedback (Jaech et al., 2024; Shao et al., 2024), producing high-performing reasoning systems (e.g., OpenAI's o1 and DeepSeek-R1; Jaech et al., 2024; Guo et al., 2025).

Numerous experimental and theoretical works have shown the benefits of CoT in improving transformers' generalization (Wei et al., 2022b; Yao et al., 2023) and their expressive power (Merrill & Sabharwal, 2023; Li et al., 2024), respectively. While recent studies show that augmenting the training data with CoT-guided data provides better learning signals (Lightman et al., 2023; Hsieh et al., 2023; Guo et al., 2025), to the best of our knowledge, understanding the learning behavior of CoT-guided transformers is still missing. To study "how

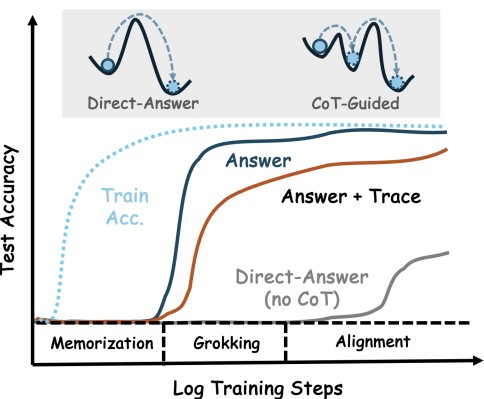

Figure 1: CoT as a learning catalyst to accelerate *grokking*, where the test accuracy follows a predictable logistic function. With CoT, test accuracy groks earlier and reaches a higher ceiling; Answer outpaces Answer+Trace (the unfaithfulness gap) before aligning, while Direct-Answer groks later. Inset: the memorization-to-generalization barrier, as an energy-landscape sketch, showing CoT splitting a hard task into smaller ones.

transformers learn to reason with CoT?", we focus on the *grokking* perspective (Power et al., 2022; Varma et al., 2023) of transformers, where a *sudden* transition from memorization to generalization occurs.

We introduce a framework under controlled experiments (Sec 3) to study and isolate the learning dynamics of CoT. We train from-scratch transformers on a suite of formal reasoning tasks including *(i)* COMPARISON, *(ii)* SORTING, *(iii)* INTERSECTION, *and (iv)* COMPOSITION with controllable algorithmic complexity based on a synthetic knowledge base. This setup allows the direct comparison of a answer-only baseline to a model trained to emit a step-by-step trace before the answer, quantifying the impact of CoT on learning dynamics. We structure the study around four research questions (RQs):

1. **RQ1: Expressivity and limits.** How does CoT improve the expressivity of transformers and how is this improvement reflected in the reasoning accuracy?

2. **RQ2: Learning dynamics.** How does CoT change learning dynamics during training with respect to rate of learning, data composition, and maximal accuracy?

3. **RQ3: Faithfulness.** Do generated traces causally mediate answers, or do models often answer correctly while ignoring the CoT traces?

4. **RQ4: Reasoning mechanism.** How does the CoT-guided transformer answer the same input query, comparing to an answer-only baseline?

We demonstrate that CoT can improve transformer generalization on solvable tasks including linear time COMPARISON and logarithmic time SORTING, and increase the expressivity of the transformer to solve a linear-time sequential COMPOSITION task (RQ1, Sec 4). Though, such improvements are contingent upon the task complexity where a CoT-guided transformer fails on a bilinear-time INTERSECTION task. Regarding *grokking* of transformers, our empirical studies reveal (Figure 1):

- Both CoT and non-CoT training paradigms can be described by predictable logistic curves (see Figure 1), determined by a three-parameter function (Sec 5). These parameters are quantitatively determined by task complexity, data distribution, and the presence of CoT. Comparing the two curves (gray and dark blue lines), **CoT acts as a catalyst** that exponentially accelerates generalization by splitting a hard task into smaller ones (RQ2, Sec 5).

- Early at training, there is a *trace unfaithfulness* transient state, before the model later aligns with its CoT reasoning (see Figure 1, brown line); (RQ3, Sec 6).

Additionally, mechanistic studies confirm that this CoT catalytic effect stems from altering the internal representation and causal correspondence among tokens (RQ4, Sec 7). Overall, our experimental setting studies the **learning process** of CoT transformers in the lens of grokking. Our results show that augmenting pretraining data with CoT supervision can substantially accelerate transformer generalization in certain tasks. However, prolonged training is suggested to align the generated trace and answer of CoT-based transformers. We caution against using generated traces as an explainable thinking process.

## 2 RELATED WORKS

**Measuring (CoT) transformer generalization.** We present a synthetic suite of reasoning tasks to measure how transformers generalize via CoT training. Similar to prior works (Power et al., 2022; Liu et al., 2022; Varma et al., 2023), such controlled experiments allow us to observe *grokking*, where the sudden transition from memorization to generalization occurs. Related to our work, Wang et al. (2024); Abramov et al. (2025); Ye et al. (2025) observe that transformers struggle to generalize on compositional data. In contrast, we show that, in the presence of CoT data, this generalization is achieved. Other works measure (CoT) transformer generalization to out-of-distribution (OOD) tasks (Dziri et al., 2023; Thomm et al., 2024; Lin et al., 2025; Zhao et al., 2025), showing their limitations. We focus on OOD generalization within the same task, since the methods to achieve task-level generalization is still under active exploration (Sanh et al., 2021; Lampinen et al., 2025).

**Learning curves of transformers.** In this work, we study the learning process of transformers with respect to generalization in the presence of CoT data. Related works have studied learning

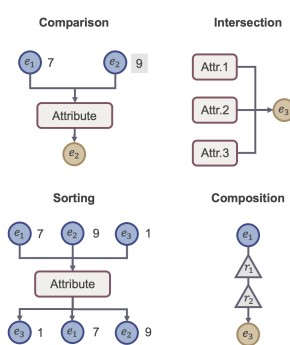

Figure 2: Illustration of the four formal symbolic reasoning tasks investigated.

Table 1: **Reasoning Tasks.** (a) Formal spec; (b) verbalized examples with tokens from $\mathcal{E}/\mathcal{A}/\mathcal{R}/\mathcal{V}$.$^\star$ Complexity of deterministic algorithms.

| Task | KB Type | Ground-truth $f(q)$ | Complexity $\mathcal{O}^\star$ |
|---|---|---|---|
| COMPARISON | Attributive $\mathcal{KB}_{\text{attr}}$ | $\arg\max / \arg\min_e \mathcal{KB}_{\text{attr}}(e, a_j)$ | $O(k)^\dagger$ |
| SORTING | Attributive $\mathcal{KB}_{\text{attr}}$ | $\text{argsort}(\mathcal{KB}_{\text{attr}}(e_i, a_j))$ | $O(k \log k)^\ddagger$ |
| INTERSECTION | Attributive $\mathcal{KB}_{\text{attr}}$ | $\bigcap_{i=1}^{k}\{e : \mathcal{KB}_{\text{attr}}(e, a_i) = v\}$ | $\tilde{O}(k \cdot s)^\S$ |
| COMPOSITION | Relational $\mathcal{KB}_{\text{rel}}$ | $f_{r_k} \circ \cdots \circ f_{r_1}(e_h)$ | $O(k)$ |

$^\dagger$ unique argmax/min. $^\ddagger$ all values distinct (total order). $^\S$ unique solution; $s$ = avg. candidate set size.

| Task | Example & Tokens |
|---|---|
| COMPARISON | *Example:* "max(height) ? Alice, Bob, Chloe" → Bob
*Tokens:* $E = \{\text{Alice, Bob, Chloe}\}$, $A = \{\text{height}\}$, $V = \{65, 72, 68\}$ |
| SORTING | *Example:* "sort height ? Alice, Bob, Chloe" → (Chloe, Alice, Bob)
*Tokens:* $E = \{\text{Alice, Bob, Chloe}\}$, $A = \{\text{height}\}$, $V = \{65, 72, 68\}$ |
| INTERSECTION | *Example:* "find $e$ with $(a_1, a_2, a_3)$ all = +1" → $e^\star$
*Tokens:* $E = \{e^\star, \ldots\}$, $A = \{a_1, a_2, a_3\}$, $V = \{+1\}$ |
| COMPOSITION | *Example:* "start: Alice; works_at → hq_in → country_of" → USA
*Tokens:* $E = \{\text{Alice, Company, City, USA}\}$, $R = \{\text{works_at, hq_in, country_of}\}$ |

curves with respect to (i) dataset and model size (Kaplan et al., 2020; Hoffmann et al., 2022; Wei et al., 2022a), as well as data distribution (Muennighoff et al., 2023; Shukor et al., 2025), and (ii) memorization and the presence of factual knowledge (Tirumala et al., 2022; Lu et al., 2024; Chang et al., 2024; Allen-Zhu & Li, 2024; Morris et al., 2025). Different from those insights, we show how the data complexity and the presence of CoT affect generalization process.

**Unfaithfulness of CoT explanations.** Although CoT is often presented as an explanation, multiple causal studies report that generated traces do not need to mediate predictions: Delete, permute, or contradict steps can leave answers unchanged, and mediation analyzes report weak or inconsistent reliance on generated steps (Turpin et al., 2023; Lanham et al., 2023; Paul et al., 2024). These results caution against interpreting CoT as the model's computation without additional evidence. Lyu et al. (2023) uses external deterministic solvers to execute machine-translated queries to bypass unfaithful reasoning chains to derive answers.

## 3 PROBLEM DEFINITION AND NOTIONS

To build a controlled environment that separates **reasoning** from **memorization**, we use a synthetic knowledge base (KB) that is either attributive or relational, from which we derive two types of data for training and testing: (i) **atomic facts** ($\mathcal{F}$) and (ii) **composed facts** ($\mathcal{D}$). The KB contains a fixed set of atomic facts, denoted $\mathcal{F}_{\text{base}}$, where each fact is a simple relational or attributive triplet (e.g., (subject, relation, object)). The composed facts $\mathcal{D}$ are generated from this base to serve as reasoning tasks, where each example requires combining multiple atomic facts from $\mathcal{F}_{\text{base}}$. We systematically control the difficulty of these reasoning tasks using two key parameters: (1) **Complexity** $k$ determines the number of atomic facts to be combined for composed facts (2) **Data Ratio** ($\phi$) determines the training set ($\mathcal{D}_{\text{train}}$) size relative to the all atomic facts in the KB, $\phi = \frac{|\mathcal{D}_{\text{train}}|}{|\mathcal{F}_{\text{base}}|}$.

### 3.1 SYNTHETIC ENVIRONMENT AND REASONING TASKS

Let $\mathcal{T}$ denote the token vocabulary, decomposed as

$$\mathcal{T} = \underbrace{\mathcal{E}}_{\text{entities}} \cup \underbrace{\mathcal{A}}_{\text{attributes}} \cup \underbrace{\mathcal{R}}_{\text{relations}} \cup \underbrace{\mathcal{V}}_{\text{values}}$$

Here $\mathcal{E} = \{e_0, \ldots, e_{N-1}\}$ is a finite set of entities, $\mathcal{A} = \{a_0, \ldots, a_{M-1}\}$ attributes, $\mathcal{R} = \{r_0, \ldots, r_{P-1}\}$ relation labels, and $\mathcal{V} \subset \mathbb{Z}$ scalar values. We write $\langle \cdot \rangle$ for linearized token sequences. We use two knowledge bases (KBs $\mathcal{KB} \in [\mathcal{KB}_{\text{attr}}, \mathcal{KB}_{\text{rel}}]$) that ground all queries: **Attributive KB** $\mathcal{KB}_{\text{attr}} : \mathcal{E} \times \mathcal{A} \to \mathcal{V}$ maps an entity–attribute pair to a value. An atomic fact is a triple $(e, a, v)$ with $v = \mathcal{KB}_{\text{attr}}(e, a)$; **Relational KB** $\mathcal{KB}_{\text{rel}}$ induces a labeled directed multigraph $G = (\mathcal{E}, \mathcal{L})$ with $\mathcal{L} \subseteq \mathcal{E} \times \mathcal{R} \times \mathcal{E}$. An atomic fact is a triple $(e_h, r, e_t) \in \mathcal{L}$.

As shown in Figure 2, we study COMPARISON, SORTING and INTERSECTION based on a $\mathcal{KB}_{\text{attr}}$. COMPARISON takes $k$ entities and one attribute then returns the unique entity with the max./min.

value. SORTING returns the same ranked $k$ entities by their attribute values. INTERSECTION takes $k$ attributes as conditions and a target value $v$ and then returns the unique entity that satisfying all conditions. Based on a $\mathcal{KB}_{\text{rel}}$, COMPOSITION takes a head entity and a $k$-relation path and returns the tail entity via sequential lookup. Formal definitions and ideal algorithmic complexities are summarized in Table 1 with verbalized examples.

## 3.2 DATA SYNTHESIS AND SPLITS

**Data Synthesis.** Our goal is to test if a transformer can learn abstract reasoning patterns and apply them to entities it has not encountered in training composed examples $\mathcal{D}_{\text{train}}$. To do this, we first partition the total entity set $\mathcal{E}$ into two disjoint subsets: an in-distribution (ID) set ($\mathcal{E}_{\text{ID}}$) and an out-of-distribution (OOD) set ($\mathcal{E}_{\text{OOD}}$).

The training data is composed of two parts: (1) the model receives the complete set of atomic facts ($\mathcal{F}_{\text{base}}$, which covers all entities from both $\mathcal{E}_{\text{ID}}$ and $\mathcal{E}_{\text{OOD}}$; (2) the model is also given a set of multi-step composed facts ($\overline{\mathcal{D}_{\text{train}}}$) that are constructed from entities exclusively from the ID set, $\mathcal{E}_{\text{ID}}$. The test set ($\mathcal{D}_{\text{test}}$) then consists of new composed facts using only entities from the held-out $\mathcal{E}_{\text{OOD}}$. This forces the model to generalize the learned abstract reasoning patterns on $\mathcal{E}_{\text{ID}}$ to $\mathcal{E}_{\text{OOD}}$. In our experiments, the data ratio ($\phi = \frac{|\mathcal{D}_{\text{train}}|}{|\mathcal{F}_{\text{base}}|}$) typically varies between 3.6 and 12.6.

**Query Synthesis.** From our constructed KBs, we synthesize the queries $q$ for $\mathcal{D}_{\text{train}}$ and $\mathcal{D}_{\text{test}}$ according to the **complexity parameter** ($k$), which sets the scale of the task (e.g., number of entities to compare, attribute conditions, or compositional hops). As defined previously, queries in the training set ($\mathcal{D}_{\text{train}}$) are **ID**, constructed exclusively with entities from $\mathcal{E}_{\text{ID}}$. Vice versa, queries in the test set ($\mathcal{D}_{\text{test}}$) are **OOD** by only using entities from $\mathcal{E}_{\text{OOD}}$. To ensure each query has a unique and unambiguous answer, we apply task-specific constraints, such as discarding samples with tied values for COMPARISON tasks. The complete generation protocols are detailed in Appendix A.

## 3.3 OUTPUT FORMATS AND EVALUATION METRICS

To isolate the impact of CoT on learning, we compare two supervision methods. For any given task, we train two separate instances of the same model architecture (fixed number of layers, number of parameters, etc.) that differ only the format of the target sequences $Y$ they are trained to predict. Given a query $q$ with a ground-truth reasoning trace $y_{\text{trace}}$ and final answer $y_{\text{ans}}$, we train an autoregressive model $p_\theta$ by maximizing the likelihood of the target $Y$.

**Direct-Answering (Non-CoT Baseline).** In this setting, the model is supervised to produce only the final answer, requiring it to perform all reasoning steps internally. The training objective is to maximize the conditional log-likelihood of the answer token:
$$\mathcal{L}_{\text{direct}} \;=\; \log p_\theta(y_{\text{ans}} \mid q) \,, \tag{1}$$

**CoT-Guided Generation.** In this model, the model is trained first generate the explicit reasoning trace and then the final answer. The target sequence $Y$ is the concatenation of both parts, where $Y = \langle y_{\text{trace}}, y_{\text{ans}} \rangle$. Using the chain rule, the objective is:
$$\mathcal{L}_{\text{cot}} \;=\; \log p_\theta(Y \mid q) \;=\; \log p_\theta(y_{\text{trace}} \mid q) + \log p_\theta(y_{\text{ans}} \mid y_{\text{trace}}, q) \,, \tag{2}$$

During training, the model learns to predict each token in the sequence $Y$ including the intermediate reasoning trace before the answer. As conceptualize in Figure 1, this method of supervising the full sequence encourages the model to break down a difficult task into a series of simpler steps.

**Metrics.** Since we know both the answer and intermediate steps from our constructed KBs, we report Final Answer, Trace (Intermediate), and Full Sequence accuracies. Final Answer accuracy measures the correctness of answer token regardless of the trace[1] Trace accuracy measures whether the generated intermediate steps match the provided trace *in-order*. Full Sequence accuracy requires both answer and trace to be correct simultaneously, respectively. We report the hold-out OOD test accuracies in this work. When evaluating Full Sequence accuracy, we require generated traces to follow the query-specified correct order, and all composed training examples adhere to the same convention.

---

[1]For SORTING, we report a token-wise partial score since there are more than one answer entity tokens.

## 3.4 Experimental Design

**Knowledge Base Construction and Split.** For tasks based on the $\mathcal{KB}_{attr}$, we use a knowledge base containing $|\mathcal{E}| = 1000$ entities and $|\mathcal{A}| = 20$ attributes. For the COMPARISON task, attribute values are sampled from the range $v \in [0, 20]$. To reduce value collisions and ensure unique answers, we increase the value range to $v \in [0, 100]$ for SORTING. For more complex INTERSECTION, we use $|\mathcal{A}| = 100$ attributes with a value range of $v \in [0, 50]$. For the COMPOSITION task, we construct a separate Relational KB ($\mathcal{KB}_{rel}$) with $|\mathcal{E}| = 1000$ entities and $|\mathcal{A}| = 20$ attributes. Following the procedure described in Sec 3.2, we partition the entity set $\mathcal{E}$ into ID and OOD sets using a 90/10 ratio. The training and test sets ($\mathcal{D}_{train}$ and $\mathcal{D}_{test}$) are then generated based on this split. To prevent the KB scale acting as a confounding factor, we use a fixed-size KB for each task family to isolate the effect of task complexity ($k$) and data ratio ($\phi$) on learning dynamics.

**Model and Tokenization.** For all experiments, we use a 12-layer GPT-2 style transformer with a 768-dimensional hidden state. Each entity, attribute, relations, and value mapped to a unique token (e.g. $\langle e_1 \rangle$, $\langle attr_1 \rangle$, $\langle 1 \rangle$) as described in Sec 3.1. The final input sequences are linearized according to the supervision format (Direct-Answering or CoT-Guided), with templates provided in Appendix A. All models were trained for up to 200k steps using the AdamW optimizer (Loshchilov & Hutter, 2017) with a mini-batch size of 256. Experiments were conducted on an 8 A100 GPU machine.

## 4 CoT Reasoning Improves Generalization

We now present our main findings with an analysis of how CoT supervision affects the model generalization on OOD performance after extensive training (200k steps). As summarized in Table 2, training with CoT consistently improves results compared to the direct-answering baseline.

**CoT Enhances Generalization and Expressivity** For attributive tasks like COMPARISON and SORTING, CoT-guided models consistently outperform their non-CoT (direct-answering) counterparts. The benefit of CoT becomes more obvious as task complexity increases. For instance, in the SORTING task ($k = 3$), the CoT model achieves 92% OOD

Table 2: Final Answer accuracy and *in-order* Intermediate accuracy (for CoT-guided models; Sec 3.3) on the full suite of reasoning tasks[†].

| | | Answer Acc. | | | | Intermediate Acc. | |
| | | CoT | | non-CoT | | CoT | |
| **Task** | $k$ | ID | **OOD** | ID | **OOD** | ID | **OOD** |
|---|---|---|---|---|---|---|---|
| Comparison | 3 | 1.00 | 0.96 | 1.00 | 0.91 | 1.00 | 0.95 |
| | 4 | 1.00 | 0.91 | 1.00 | 0.86 | 1.00 | 0.91 |
| Sorting | 3 | 1.00 | 0.92 | 1.00 | 0.18 | 1.00 | 0.99 |
| | 4 | 1.00 | 0.83 | 1.00 | 0.04 | 1.00 | 0.98 |
| Composition | 2 | 1.00 | 1.00 | 1.00 | 0.00 | 1.00 | 1.00 |
| | 3 | 1.00 | 1.00 | 1.00 | 0.00 | 1.00 | 1.00 |
| Intersect | 2 | 1.00 | 0.01 | 1.00 | 0.00 | 1.00 | 0.93 |
| | 3 | 1.00 | 0.07 | 1.00 | 0.00 | 1.00 | 0.84 |

[†]Accuracies are reported with $\phi = 12.6$. We denote higher $\phi$ means more composed reasoning data for the model to learn from (Sec 3.2). We report more results in Appendix C

accuracy where the non-CoT model fails to generalize at 18%. Meanwhile, CoT-guided models also achieve near-perfect intermediate accuracy predicting the corresponding values. However, we note these improvements are contingent upon the model architecture: a Mamba (Gu & Dao, 2023) model with matched parameters fails to generalize on COMPARISON task even with the same CoT, indicating that the inductive biases of the Transformer architecture are crucial to utilize CoT (Appendix C.3).

The most obvious effect is observed in the COMPOSITION task, which requires inherently sequential lookups. Here, the non-CoT model completely fails to generalize, while the CoT-guided model achieve strong OOD generalization within 3k steps (Appendix C.5). From Table 2, we also observe the CoT-guided model reach perfect intermediate accuracy in composing bridge entities. This finding demonstrates that CoT expands the expressivity of a transformer to solve sequential problems.

**A Practical Computational Frontier.** Despite its benefits for tasks like COMPOSITION, our results also reveal a practical computational frontier where the CoT supervision fails to improve generalization. This limit is shown in the model's consistent failure on the INTERSECTION task. Algorithmically, this tasks is significantly more demanding than the others, as its solution requires maintaining and intersection $k$ different sets of candidate entities.

We find both training paradigms failed to generalize on OOD data for this task, even with various CoT templates. A closer analysis of the intermediate steps reveals the specific failure: while the model could often correctly predict the candidate lists for individual conditions, it consistently failed at identifying the single entity common to all $k$ sets (as detailed in Appendix C.4).

# 5 AUTOCATALYTIC PHASE TRANSITION IN (COT-GUIDED) TRANSFORMERS

In this section, we analyze the learning dynamics by tracking the OOD accuracy, $Acc(t)$, over logarithmic training steps. We propose a kinetic model, consisting of first- and second-order principles, to quantitatively describe the learning curves for both the CoT and non-CoT paradigms. As the results will demonstrate, this model accurately approximates the learning dynamics of transformers on tasks such as COMPARISON and SORTING. We show that CoT acts as a catalyst, lowering the barrier of learning reaction (kinetics).

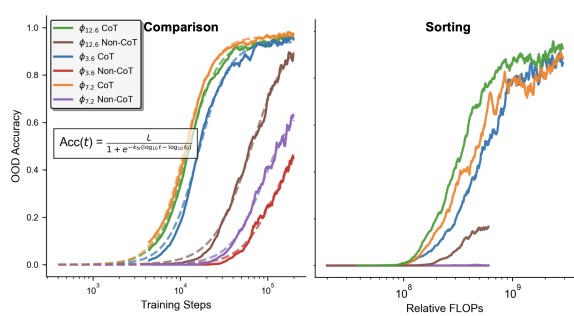

Figure 3: Analysis of model accuracy over (left) log training steps for the $k = 3$ COMPARISON task (right) log training FLOPs the $k = 3$ SORTING task. **Solid lines** represent the averaged experimental results, while **dashed lines** show the fitted theoretical functions.

## 5.1 GENERALIZATION DYNAMICS

As conceptually visualized in Figure 1, *grokking* follows a sharp sigmoidal curve similar to a phase transition. This empirical behavior, shown in the learning curves of COMPARISON and SORTING (Figure 3), which we later explain, is well described by an autocatalytic picture (Moore & Pearson, 1981): once a small seed of correct solution appears, it catalyzes its own growth. We introduce a kinetic model to quantitatively describe theses behaviors, which consists of (i) a first-order dynamic function for describing a learning curve, (ii) and second-order effects that govern how these curves change with task and training conditions.

**The First-Order Function: Modeling the Learning Phase Transition**  Let $Acc(t)$ denote OOD accuracy at training step $t$. A classic autocatalysis model in linear step is logistic,

$$\frac{dAcc}{dt} = r \cdot Acc\,(L - Acc),\tag{3}$$

with maximum accuracy $L$ and linear-time growth rate $r$. Empirically, however, our learning curves are fit more parsimoniously by a single logistic curve in the logarithm of steps:

$$Acc(t) = \frac{L}{1 + \exp\Big(-k_{\text{fit}}\big(\log_{10} t - \log_{10} t_0\big)\Big)}.\tag{4}$$

Here $t_0$ is the geometric take-off point (the step where $Acc = L/2$), and $k_{\text{fit}}$ is the slope of the fitted function in log-step. This three-parameter kinetic model serves as our first-order dynamic function to describe how OOD accuracy scales with the training steps $t$ for one run, allowing comparison across tasks and supervision modes via $(L, t_0, k_{\text{fit}})$. We show our simple fitting procedure using `Scipy` (Virtanen et al., 2020) in Appendix B, where we fitted the accuracy metrics in logarithm of training steps to Eq. 4. Figure 3 illustrates both the accuracy of our kinetic model and its utility for analysis. The left panel demonstrates that our logistic function (dashed lines) provides an excellent fit for the empirical learning curves (solid lines) on the COMPARISON task. The right panel, which plots accuracy against relative FLOPs[2] for the SORTING task, shows that CoT-guided models are far more compute-efficient. This efficiency is illustrated in the SORTING task, where CoT-guided models achieve high accuracy using $10^9$ FLOPs and the non-CoT baselines fail.

## 5.2 THE SECOND-ORDER LAWS AND THEIR ARRHENIUS INTERPRETATION

---

[2]We use *relative* FLOPs by estimating the total tokens since we use a fixed-size model.

Table 3: Fitted kinetics for SORTING across data ratios ($\phi$) and task complexity ($k$). Increasing $k$ delays take-off ($t_0$), reduces the normalized rate $\hat{r}$ and ceiling $L$, and increases the steepness $k_{\text{fit}}$. Reported for each fit: $R^2$ and RMSE.

| Task Params. | | Fitted Curve Params. (Eq. 4) | | | Resulted Rate (Eq. 6) | Fit Metrics | |
|---|---|---|---|---|---|---|---|
| $\phi$ | $k$ | $L$ | $k_{\text{fit}}$ | $t_0$ | $\hat{r}$ | $R^2$ | $RMSE$ |
| | 3 | 0.887 | 5.67 | 86K | $3.2 \times 10^{-5}$ | 0.919 | 0.0778 |
| 3.6 | 4 | 0.842 | 7.45 | 148K | $2.6 \times 10^{-5}$ | 0.923 | 0.0852 |
| | 5 | 0.738 | 9.97 | 200K | $2.9 \times 10^{-5}$ | 0.904 | 0.0936 |
| | 3 | 0.851 | 5.33 | 66K | $4.1 \times 10^{-5}$ | 0.879 | 0.0866 |
| 7.2 | 4 | 0.834 | 5.04 | 121K | $2.2 \times 10^{-5}$ | 0.872 | 0.0980 |
| | 5 | 0.775 | 5.65 | 183K | $1.7 \times 10^{-5}$ | 0.856 | 0.1031 |
| | 3 | 0.906 | 6.45 | 55K | $5.7 \times 10^{-5}$ | 0.932 | 0.0650 |
| 12.6 | 4 | 0.862 | 6.36 | 85K | $3.8 \times 10^{-5}$ | 0.909 | 0.0836 |
| | 5 | 0.795 | 7.02 | 118K | $3.2 \times 10^{-5}$ | 0.882 | 0.0988 |

While the first-order function describes a single learning curve, we now turn to the second-order principles that govern it, and how the parameters of that learning curve ($L, t_0, k_{\text{fit}}$) change based on experimental conditions. We find these parameters vary systematically with experimental conditions, such as task complexity $k$, data ratio $\phi$, and the presence of CoT.

To unify these second-order effects, we introduce a conceptual framework analogous to the **Arrhenius equation** (Arrhenius, 1889; Laidler, 1984) from chemical kinetics. This model posits that the rate of learning determined by two factors: (i) an **activation barrier** ($\Delta$) which increases with task complexity ($k$), and (ii) an **effective temperature** ($T_{\text{eff}}$) which increases with the data ratio ($\phi$). Based on the two-part structure of the CoT loss function (Eq. 2), we propose the overall learning rate where *grokking* happens (Eq. 3) depends on parallel pathways for predicting the trace and answer:

$$r \; \propto \; \exp\Big( - \Delta_{\text{trace}}(k)/T_{\text{eff}}(\phi)\Big) + \exp\Big( - \Delta_{\text{ans}|\text{trace}}(k)/T_{\text{eff}}(\phi)\Big), \tag{5}$$

Arrhenius equation decides the shape of learning dynamics via two main factors: the barrier $\Delta$ grows with task difficulty (e.g., larger $k$), while the effective temperature $T_{\text{eff}}$ increases with data distribution (e.g., higher data ratio $\phi$). Here we relate the rate to two smaller barriers per Eq. 2, as CoT guidance introduces intermediate tokens $\Delta_{trace}$). Specifically, Table 3 reports fitted parameters and linear growth rate $r$ from the fitted slope $k_{\text{fit}}$, where we approximate $r$ normalized by the maximal accuracy $L$ as

$$\hat{r} = k_{\text{fit}}/(t_0 L \ln 10). \tag{6}$$

This approximation connects the fit slope $k_{\text{fit}}$ to the normalized rate $\hat{r}$. This rate is the quantity of our Arrhe-

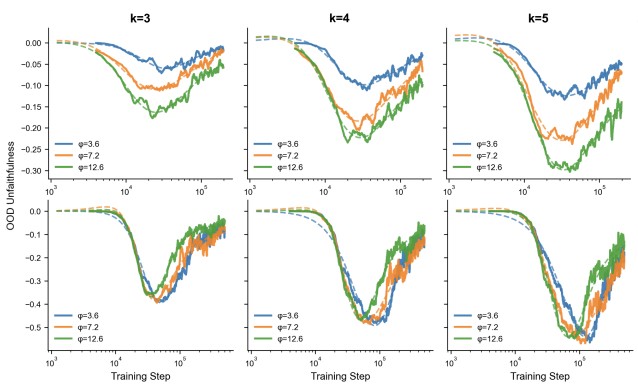

Figure 4: Unfaithfulness dynamics of the COMPARISON (Top) and SORTING (Bottom) tasks with $k = 3, 4, 5$. For SORTING task we use the final answer accuracy assigning partial credits. Unfaithfulness follows a reverse double descent pattern, which is deeper for SORTING task.

nius analogy (Eq. 5), allowing us to empirically verify the framework using the fitted parameters. As Table 3 shows, as difficulty increases from $k$=3 to $k$=5 at fixed $\phi$=3.6, the take-off point is delayed ($t_0: \; 86K \rightarrow 200K$) with lower linear rate ($\hat{r}: 3.2e^{-5} \rightarrow 2.9e^{-5}$), while the fitted slope increases ($k_{\text{fit}}: \; 5.67 \rightarrow 9.97$). Under this conceptual model, CoT acts as a catalyst. By providing intermediate reasoning steps (Eq. 2), it effectively lowers the learning difficulty, which is represented by the barrier $\Delta$. A linear reduction in the *difficulty barrier* can lead to an exponential decrease in the training steps required to generalize. Our Arrhenius analogy (Eq. 5) predicts that the learning rate $r$ increases with the data ratio $\phi$, and from Table 3, we generally see this trend hold (e.g, $\hat{r}$ increases from $3.2e^{-5}$ to $5.7e^{-5}$ as $\phi$ increases from 3.6 to 12.6), but with minor fluctuation for more complex tasks, which likely stem from the experimental variance and noise in the training process.

## 6 UNFAITHFULNESS IN THE COT REASONING

While CoT supervision improves final performance, a rising question (RQ3) is whether the generated traces are **faithful**, which refers to the alignment between the generated intermediate trace and final answer. This section analyzes the phenomenon of "unfaithfulness," which occurs when the model predicts the correct final answer despite having a wrong or contradicting trace.

To quantify this, we define the **unfaithfulness gap** as the difference between the Final Answer accuracy and the Full Sequence accuracy following procedure described in Sec. 3.2. A larger value for this gap indicates a higher degree of unfaithfulness, where the model produces correct answers with contradicting reasoning trace. This metric is plotted over the training steps in Figure 4. The two individual learning curves (Full Sequence and Final Answer accuracies are each well fit by the kinetic model from Section 5. By analyzing the evolution of this gap, we can understand unfaithfulness not as a simple failure pattern, but as a transient phase in the learning dynamics.

**Empirical pattern.** As show in Figure 4 across COMPARISON and SORTING, the unfaithfulness curve follows a characteristic three-phase trajectory. Early in training, the gap is near zero (the model has learned neither answers nor procedure). In the intermediate phase, the gap opens and peaks, indicating that the model has found a shortcut for answers before it follows the given CoT program. In late training, the gap closes as traces and answers align. The peak size grows and alignment is delayed as tasks become harder (larger $k$ or longer paths), while increasing $\phi$ increases the gap in the COMPARISON task. In fact, the intermediate accuracy increases faster than the answer accuracies (Appendix C.2) for both COMPARISON and SORTING tasks. This suggests the model finds non-faithful shortcuts to the answer even when it has the necessary knowledge for a faithful solution.

**Implications.** Unfaithful reasoning is not merely a failure mode but a transient learning phase that is sensitive to both task complexity and data composition. This cautions against single-pass training implied by scaling-law heuristics: without sufficient optimization, models may plateau in the shortcut basin—producing correct answers with flawed or hallucinatory traces. This finding also cautions hallucination detection and explanatory methods depending on intermediate reasoning process can make mistakes.

## 7 COMPUTATIONAL MECHANISMS IN (CoT-GUIDED) TRANSFORMERS

Having established the high-level learning dynamics, this section investigates the internal computational mechanism the model learns to perform reasoning (RQ4). We aim to answer two concrete questions: (1) Where in the transformer are the features necessary for solving the reasoning task formed? (2) Which of these internal representations are causally responsible for deriving the final answer? To address both questions, we employ two complementary analysis techniques: linear probing (Alain & Bengio, 2016)

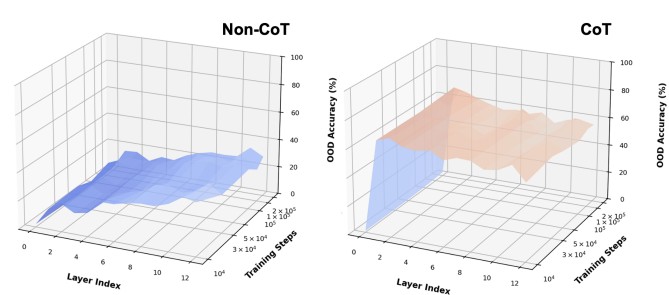

Figure 5: Answer-probe accuracy (z-axis) on the OOD data in COMPARISON, plotted as a surface over model layers (x-axis) and training steps (y-axis).

and causal tracing (Meng et al., 2022). Linear probes determine if the task-relevant features is decodable from a layer's hidden states, while the causal tracing identifies critical states for the predicted answer. Full protocols are detailed in Appendix D.

### 7.1 LINEAR PROBING

We use linear probing to investigate where the final answer in computed within the transformer using two training paradigms. Figure 5 visualizes the probing results as two surface plots, one for the non-CoT model (left) and one for the CoT-guided model (right). In each plot, the surface shows the accuracy of a probe trained to predict the final answer. The two horizontal axes represent the layer depth and progression of training steps, and the vertical axis represents the probing accuracy on the OOD split. A high point on the surface indicates that the final answer is easily decodable from the hidden representations at that specific layer. For a non-CoT model, the final answer is not decodable in the early layers, rises through the middle layers, and peaks near the output layer. This surface with a gradual raise as the layer depth increases indicates the non-CoT model learned a gradual, layer-by-layer computation. In contrast, the probing accuracy for a CoT-guided model

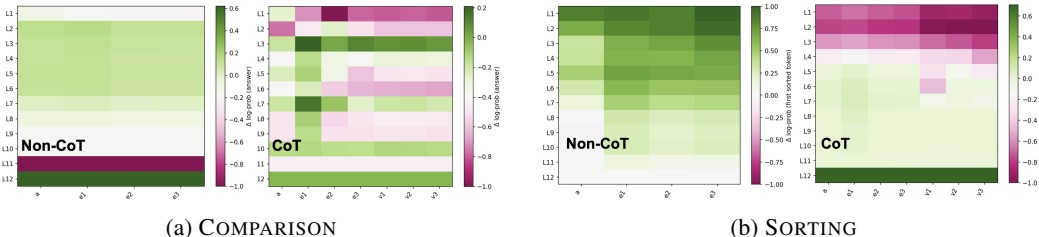

(a) COMPARISON  (b) SORTING

Figure 6: Causal tracing heatmaps on the OOD split for the COMPARISON (a) and SORTING (b) tasks. The heatmaps contrast the computational pathways learned by the non-CoT (left of each pair) and CoT-guided (right) models. Color indicates the causal importance, where **green** denotes a critical state to the answer. On the vertical axis, a denotes the attribute, e denotes the entities, and v denotes the corresponding values in the trace.

remains high across the entire depth of the transformer. This indicates that once the model generates the reasoning trace and includes it in the context, the problem is fundamentally simplified.

## 7.2 CAUSAL TRACING

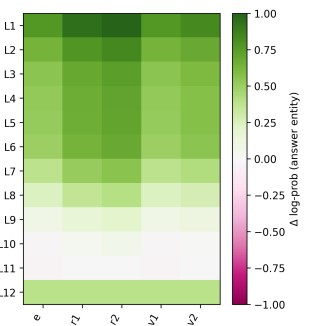

Figure 7: Causal Tracing results of CoT-guided model on the two-hop COMPOSITION. $e$ denotes the head entity, $r$ denotes the relations, and $v$ denotes entities in the trace.

While linear probes show where information is present, they do not prove causality. To identify the representations that are critically responsible for the predicted answer, we turn to causal tracing. Figure 6 visualizes these results as heatmaps for the COMPARISON and SORTING tasks. In each heatmap, the color indicates the causal importance of a token's representation at a specific model layer. A strong positive value (green) means that representation was essential for producing the correct answer. For both tasks, the non-CoT models (left panels in Fig. 6a and 6b) exhibit a clear pattern. Causal responsibility for the answer is concentrated in the early layers (layers 1-7) and fading in deeper layers, as the final answer is more decodable (Figure 5). In contrast, CoT-guided models show no clear causal correspondence between tokens and the final answer (right panels).

We demonstrate the causal tracing results of the CoT-guided model in Figure 7, as the non-CoT model fails on this task. This inherently sequential COMPOSITION task provides an important test case. The model's computational pattern is similar to the non-CoT models on other tasks, showing a strong, serial causal correspondence in the middle and upper layers of the network. This finding suggests CoT guides the model to adopt the correct computational structure for each specific task, other than a fixed inductive bias.

## 8 CONCLUSION AND LIMITATIONS

**Conclusion.** We introduced a controlled suite of formal reasoning tasks with tunable complexity and compared non-CoT vs. CoT-supervised training from scratch. Learning curves exhibit an autocatalytic picture well captured by a single logistic in log training steps, yielding compact parameters $(L, t_0, k_{\mathrm{fit}})$ that explain when and how sharply learning proceeds. CoT supervision consistently advances take-off and often raises the ceiling, but it also reveals a transient faithfulness gap—early correct answers with incorrect traces. Mechanistic analyses show that CoT shifts computation from late, serial pipelines toward earlier, more distributed representations. As task complexity scales, we observe a practical frontier: under our setups, both training paradigms gradually fail in SORTING task; and neither yields generalization on INTERSECTION.

**Limitations** The design of our study is key to isolating the learning dynamics of CoT, but our findings are subject to the following limitations: (1) our synthetic reasoning tasks can be different from the real-world reasoning tasks and our setup does not assess how natural language understanding, i.e., large-scale pretaining, affects CoT reasoning (2) we analyze a fixed-size transformer architecture that can learn the KB (Sec 3.1); extending the study to different model sizes could provide more insights on whether model scale affects the learning dynamics.

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

# A   DATASET CONSTRUCTION

## A.1   DATA CURATION FOR COMPARISON

**Vocabulary and serialization.**   We use the entity set $\mathcal{E} = \{e_0, \ldots, e_{N-1}\}$, attributes $\mathcal{A} = \{a_0, \ldots, a_{M-1}\}$, and scalar values $\mathcal{V} \subset \mathbb{Z}$, together with structural tokens $\mathcal{S}$ (e.g., q, mask, sep, eos). Linearized sequences are written with $\langle \cdot \rangle$. Each attribute $a_j$ has optional aggregator heads $\max(a_j)$ and $\min(a_j)$. An atomic attributive fact $(e, a, v)$ is serialized as $\langle e, a \rangle \mapsto \langle e, a, v, \text{eos} \rangle$.

**Attributive KB.**   We sample an integer matrix $\mathcal{KB}_{\text{attr}} : \mathcal{E} \times \mathcal{A} \to \mathcal{V}$ by drawing entries uniformly from a fixed finite range; an optional variant applies random signs to symmetrize around zero. Atomic facts enumerate all $(e, a)$ with their values via the serialization above.

**Entity split (entity-OOD).**   We split $\mathcal{E}$ uniformly at random into in-distribution (ID) and out-of-distribution (OOD) sets at a fixed ratio (we use 0.9. Atomic facts are generated for all entities and labeled by split. Inferred training examples are drawn from ID; OOD evaluation examples use only OOD entities (for MIX settings we allow both).

**Synthesis of $k$-way queries.**   For each arity $k$ and attribute $a_j$, we uniformly sample distinct $k$-tuples of entities into three pools: ID-only (all $k$ are ID), MIX (a nontrivial mix of ID and OOD), and OOD-only (all $k$ are OOD), up to a per-$(k, a_j)$ budget. Let the tuple be $(e_{i_1}, \ldots, e_{i_k})$ with values $v_\ell = \mathcal{KB}_{\text{attr}}(e_{i_\ell}, a_j)$. We discard tuples with ties in $\max$ (or $\min$) to ensure a unique target. For each valid tuple we create two examples (same retrieved $v_{1:k}$): **max** (argmax) and **min** (argmin).

**Formats.**   The input for **max** is

$$\langle \max(a_j), \text{q}, e_{i_1}, \ldots, e_{i_k}, \text{mask} \rangle,$$

and $\min(a_j)$ is used analogously for **min**. We provide two targets:

1. non-CoT: $\langle e_{i^\star}, \text{eos} \rangle$, where $i^\star = \arg\max / \arg\min_\ell v_\ell$.
2. CoT-guided: $\langle v_1, \ldots, v_k, e_{i^\star}, \text{eos} \rangle$.

**Splits and $\phi$-sweep.**   For each inferred-to-atomic ratio $\phi$, training combines all atomic facts (ID and OOD) with a subset of ID-only $k$-way examples whose size is proportional to $\phi$ times the number of ID atomic facts (capped by availability). Test uses OOD-only tuples for $k = 3, 4, 5$.

## A.2   DATA CURATION FOR SORTING

Unless noted, SORTING reuses the COMPARISON settings.

**Query synthesis (differences).**   For each $k$ and attribute $a_j$, we uniformly sample distinct $k$-tuples of entities into the same pools as COMPARISON. We discard tuples with repeated values under $a_j$ to ensure a unique total order; remaining tuples are deduplicated per $(k, a_j)$ budget.

**Serialization.**   Given a tuple $(e_{i_1}, \ldots, e_{i_k})$ with values $v_\ell = \mathcal{KB}_{\text{attr}}(e_{i_\ell}, a_j)$, the input is

$$\langle \text{sort}, a_j, \text{q}, e_{i_1}, \ldots, e_{i_k}, \text{mask} \rangle.$$

Let $\pi$ sort these values ascending. We use two targets:

1. non-CoT: $\langle e_{i_{\pi(1)}}, \ldots, e_{i_{\pi(k)}}, \text{eos} \rangle$.
2. CoT-guided: $\langle e_{i_1}, v_1, \text{sep}, \ldots, \text{sep}, e_{i_k}, v_k, \text{sep}, e_{i_{\pi(1)}}, \ldots, e_{i_{\pi(k)}}, \text{eos} \rangle$.

We use a different trace template here to further strength the canonical retrieval order specified by the input query.

**Splits.**   Train/validation/test construction mirrors COMPARISON: training uses all atomics (ID+OOD) plus a $\phi$-proportional subset of ID-only sorting tuples; validation holds out ID tuples; test prioritizes OOD-only tuples, with MIX optionally included in extended $k$ settings.

## A.3 DATA CURATION FOR INTERSECTION

**Task.** We reuse the attributive KB $\mathcal{KB}_{\text{attr}} : E \times \mathcal{A} \to V$. An INTERSECTION query specifies $k$ attribute–value conditions $(a_1{=}v_1, \ldots, a_k{=}v_k)$. The (unique) answer entity $e^\star$ must satisfy all $k$ conditions:

$$e^\star \in \bigcap_{i=1}^{k} \{e \in E : K_{\text{attr}}(e, a_i) = v_i\}, \qquad \left| \bigcap_{i=1}^{k} \{\cdot\} \right| = 1.$$

**Serialization.** We linearize inputs as $\langle \texttt{intersect}, \texttt{q}, a_1, v_1, \ldots, a_k, v_k, \texttt{mask} \rangle$. Two target formats are used: Direct (answer-only) $\langle e^\star, \texttt{eos} \rangle$; CoT-guided with variants described below.

**Query synthesis.** We first sample a provisional answer entity $e^\star \in E$ and choose $k$ distinct attributes $a_1, \ldots, a_k \in \mathcal{A}$ and values $v_1, \ldots, v_k \in V$. We "stamp" these values into $\mathcal{KB}_{\text{attr}}$ for $e^\star$, then fill remaining entries for all entities at random from $V$ subject to global marginals. We retain only queries that yield a unique satisfying entity $e^\star$. We deduplicate permutations of the same $\{(a_i, v_i)\}_{i=1}^{k}$ pattern across splits.

**Leakage prevention and splits.** We partition entities into $E_{\text{ID}}$ and $E_{\text{OOD}}$. Training always includes all atomics facts $\mathcal{AF}$ for both ID+OOD entities. Composed INTERSECTION queries used for training are constructed only from $E_{\text{ID}}$; test is OOD-only. We forbid query duplication across splits and enforce the unique-answer constraint above.

**CoT templates.** We provide multiple CoT layouts; all end with the answer.

- **R–A (retrieve→answer):** $\langle (a_1, v_1), \texttt{LIST}(a_1{=}v_1), \ldots, (a_k, v_k), \texttt{LIST}(a_k{=}v_k), e^\star, \texttt{eos} \rangle$.

- **R–C–A (retrieve→count→answer):** as above, then a `count` block summarizing entities that appear in all lists, then $e^\star$.

- **C–R–A (count→retrieve→answer):** a counting hint before retrieval lists.

## A.4 DATA CURATION FOR COMPOSITION

**Vocabulary and serialization.** We use the entity set $\mathcal{E} = \{e_0, \ldots, e_{N-1}\}$, relation labels $\mathcal{R} = \{r_0, \ldots, r_{P-1}\}$, and structural tokens from $\mathcal{S}$ (e.g., a query marker and an end-of-sequence token $\texttt{eos} \in \mathcal{S}$). Linearized sequences are written with the operator $\langle \cdot \rangle$. A $k$-hop query with head $e_h$ and relation sequence $r_{1:k}$ is serialized as $\langle e_h, r_1, \ldots, r_k \rangle$.

**Relational KB.** The relational knowledge base $\mathcal{KB}_{\text{rel}}$ induces a labeled directed multigraph $G = (\mathcal{E}, \mathcal{L})$ with $\mathcal{L} \subseteq \mathcal{E} \times \mathcal{R} \times \mathcal{E}$. For data synthesis we sample, for each $r \in \mathcal{R}$, a (near-)functional map $f_r : \mathcal{E} \to \mathcal{E}$ (implemented as a random permutation, optionally subsampled to meet an edge budget). Atomic facts enumerate the realized edges $\{(e, r, f_r(e))\}$. We precompute adjacency and all-pairs shortest-path distances used by filtering rules below.

**Entity split.** We partition $\mathcal{E}$ uniformly at random into ID and OOD sets at a fixed ratio (we use $0.9$). Atomic facts are generated for all entities and labeled by split. Inferred training examples are drawn only from ID entities; OOD evaluation examples use only OOD entities (all nodes on the path are OOD).

**Synthesis of $k$-hop queries.** For each $k$ and head $e_h$, we depth-first enumerate relation sequences $r_{1:k}$ and compute the tail $e_t = f_{r_k} \circ \cdots \circ f_{r_1}(e_h)$. To ensure difficulty and prevent leakage we enforce: (i) no shortcuts — require $\text{dist}(e_h, e_t) = k$ under the graph metric (discard cycles back to $e_h$ and any case admitting a shorter witness), and (ii) composition de-duplication — forbid reusing the same composed triple $(e_h, b_1, e_t)$, where $b_1 = f_{r_1}(e_h)$, with different relation sequences across any split. Eligible paths are bucketed by split: ID-only paths supply train/validation and OOD-only paths supply OOD test; per-$k$ budgets cap enumeration.

**Output formats.** Given a query $\langle e_h, r_1, \ldots, r_k \rangle$ with intermediates $b_1, \ldots, b_{k-1}$ and tail $e_t$, we use:

1. non-CoT: target $\langle e_t, \texttt{eos} \rangle$.
2. CoT-guided: target $\langle b_1, \ldots, b_{k-1}, e_t, \texttt{eos} \rangle$.

**Splits and $\phi$-sweep.** For each inferred-to-atomic ratio $\phi$, training combines all atomic facts (ID and OOD) with a subset of ID-only $k$-hop examples whose size is proportional to $\phi$ times the number of ID atomic facts. Test includes composed examples only from OOD.

## B    AN AUTOCATALYTIC KINETICS MODEL FOR GROKKING

The phenomenon of *grokking*, characterized by a sudden improvement in out-of-distribution generalization long after a model has achieved near-perfect in-distribution accuracy, can be analogized to a phase transition. The dynamics of this transition are well-described by an autocatalytic process, where the initial formation of a correct *seed* of neural circuitry catalyzes the rapid development of the complete, generalizable solution. This appendix details the mathematical model used to interpret the learning dynamics observed in our experiments.

---

**Algorithm 1** Logistic Curve Fitting

---

**Require:** Training step-acc. pairs $(t_i, \text{Acc}_i)$, $i = 1, \ldots, n$
**Ensure:** Parameters $(L, k_{\text{fit}}, t_0)$
1: **Curve Fitting:**
2: $x_i \leftarrow \log_{10}(t_i)$ ▷ Log-transform steps
3: $p_0 \leftarrow [\max(\text{Acc}), 1.0, \text{median}(x)]$ ▷ Initial guess
4: $(L^*, k_{\text{fit}}^*, x_0^*) \leftarrow \arg\min \sum_i [\text{Acc}_i - f(x_i)]^2$ ▷ Fit logistic
5: where $f(x) = \frac{L}{1+e^{-k(x-x_0)}}$
6: $t_0^* \leftarrow 10^{x_0^*}$ ▷ Convert back to steps

---

**The Autocatalytic Model in Linear Time**    We treat the model's accuracy, $Acc_t$ as the concentration of the *learned* state. The rate of change of accuracy is modeled by the logistic differential equation, which is characteristic of autocatalysis where a product catalyzes its own formation from a resource (the *unlearned* state), $L - Acc_t$:

$$\frac{d\text{Acc}(t)}{dt} = r \cdot \text{Acc}(t) \cdot \left( 1 - \frac{\text{Acc}(t)}{L} \right) \tag{7}$$

Here, $L$ is the maximum achievable accuracy (the saturation point, or ordered phase), and $r$ is the intrinsic growth rate of the learned structure once the network starts to generalize. The solution to this differential equation gives the accuracy as a function of linear training time $t$:

$$\text{Acc}(t) = \frac{L}{1 + e^{-r(t-t_0)}} \tag{8}$$

The parameter $t_0$ represents the midpoint of the transition, or the *takeoff time*, where the accuracy is $L/2$. It is determined by both the growth rate $r$ and the initial accuracy at the start of training, $\text{Acc}_0$:

$$t_0 = \frac{1}{r} \ln \left( \frac{L - \text{Acc}_0}{\text{Acc}_0} \right) \tag{9}$$

**Approximation for Fitting in Logarithmic Time**    Our empirical plots, such as the one shown in the main text, span several orders of magnitude of training steps. For visualization and analysis, we therefore plot accuracy against $x = \log_{10}(t)$. A direct change of variables from $t$ to $\log_{10}(t)$ in the solution above does not yield a logistic function. In practice, we observe that the accuracy curves are empirically well-approximated by a logistic function in log-time. Therefore, we fit the data using the following equation, as shown by the curves in the figures:

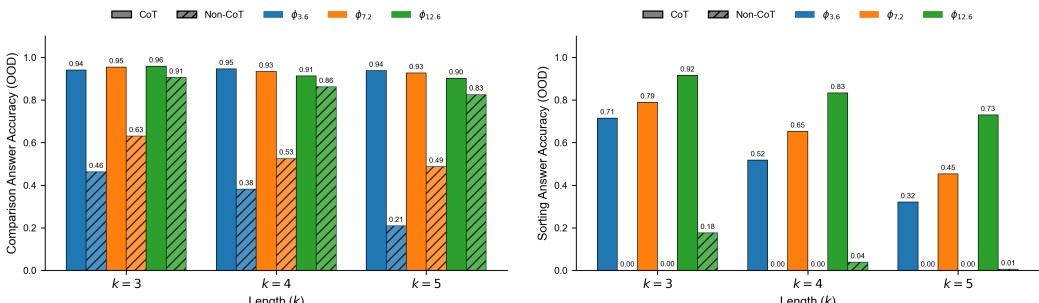

Figure 8: **Final OOD answer accuracy.** Solid bars: CoT-guided; hatched: non-CoT. Groups vary arity $k$; colors vary composed-data ratio $\phi$. **(a) COMPARISON:** CoT achieves near-ceiling accuracy across $k$ and $\phi$; non-CoT lags but improves with larger $\phi$. **(b) SORTING:** performance degrades with $k$ and improves with $\phi$; CoT helps but does not close the gap at larger $k$, while non-CoT fails for smaller $\phi$.

$$\text{Acc}(x) \approx \frac{L}{1 + e^{-k_{\text{fit}}(x - x_0)}}, \quad \text{where } x = \log_{10} t \text{ and } x_0 = \log_{10} t_0 \tag{10}$$

## C  ADDITIONAL RESULTS

### C.1  FINAL OOD ACCURACY ACROSS TASKS AND SETTINGS

Figure 8 provides a summary of the final OOD answer accuracies, comparing CoT-guided (solid bars) and non-CoT (hatched bars) models on the COMPARISON and SORTING tasks. The results are presented across different task complexities ($k$) and data ratios ($\phi$).

For the COMPARISON task (left panel), the CoT-guided models achieve near-perfect accuracy (well above 90%) across all conditions. The non-CoT models still demonstrate successful generalization that consistently improves with a higher data ratio ($\phi$).

The results for the more algorithmically complex SORTING task (right panel) are more contrastive. Here, the non-CoT models almost completely fail to generalize. In contrast, the CoT-guided models achieve strong performance, especially at lower complexities. However, the effectiveness of CoT clearly diminishes as task complexity increases from $k = 3$ to $k = 5$. For both tasks, but especially for SORTING, a higher data ratio improves the final accuracy.

### C.2  LEARNING DYNAMICS ON THE COMPARISON TASK

Figure 9 provides a detailed view of the learning dynamics for the COMPARISON task under CoT supervision, plotting three OOD accuracy metrics against training steps, task complexity ($k$), and data ratio ($\phi$). The plots reveal two primary trends. First, increasing the data ratio ($\phi$) significantly accelerates generalization, causing the Answer and Full-Sequence accuracy curves to shift to the left and indicating that the model learns in fewer steps. Conversely, increasing task complexity ($k$) makes the learning challenge greater, shifting all accuracy curves to the right. Across all settings, the Final Answer accuracy consistently rises earlier than the Full-Sequence accuracy; this gap corresponds to the "unfaithfulness" phenomenon discussed in Section 6.

### C.3  ARCHITECTURE CONTROL: STATE-SPACE (MAMBA) VS TRANSFORMER

To investigate whether the benefits of CoT supervision are a general property of sequence models or are specific to the Transformer architecture, we conducted a control experiment. We replaced the Transformer with Mamba (Gu & Dao, 2023), which is a a modern state-space model architecture. We trained a Mamba model with a parameter count matched to our Transformer on the COMPARISON task, using both direct-answering and CoT-guided supervision. As illustrated in Figure 10,

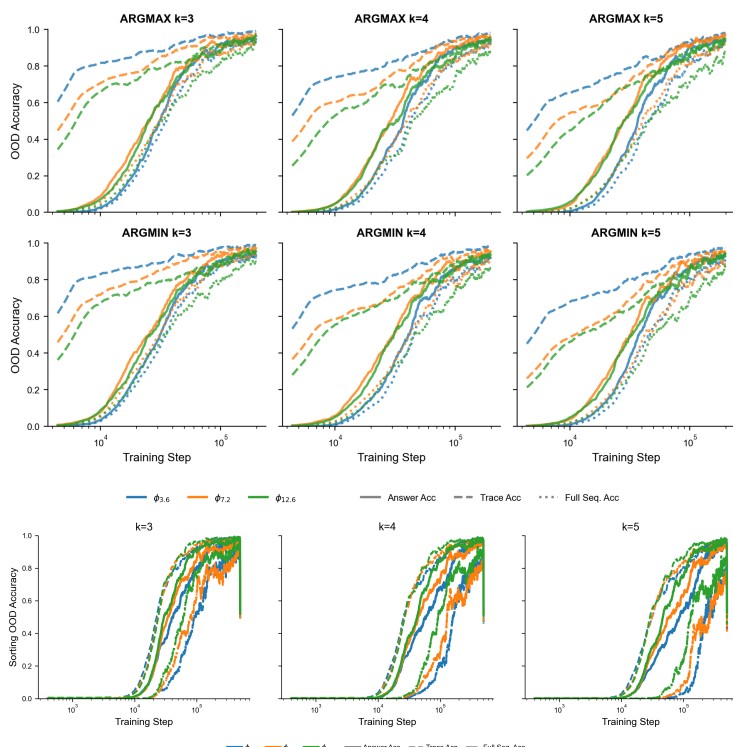

Figure 9: Training accuracy for the COMPARISON task (Top) and SORTING task (Bottom) with $k = 3, 4, 5$ (Left to Right) and data ratio $\phi = 3.6, 7.2, 12.6$ on the OOD split for a CoT-guided transformer. Solid lines represent final answer accuracy, dashed lines represent intermediate trace accuracy, and dotted lines represent full sequence accuracy, which requires both the answer and trace to be correct.

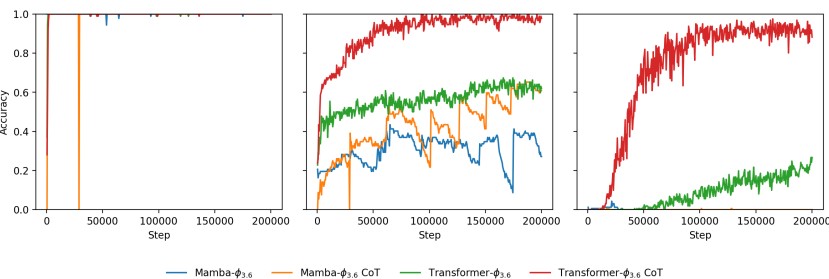

Figure 10: Training accuracy for COMPARISON task with $k = 5$ and data ratio $\phi = 3.6$ on ID, Mix, and OOD split (Left to Right) for Mamba and transformer with and without CoT guidance.

the Transformer model successfully generalizes to the OOD split, especially with CoT guidance, the Mamba model fails entirely. Mamba's OOD accuracy remains near zero under both supervision settings, even as its ID accuracy is perfect.

## C.4    ANALYSIS OF CoT TEMPLATES FOR THE INTERSECTION TASK

Given the high algorithmic complexity of the INTERSECTION task, we hypothesized that the structure of the CoT prompt might be a critical factor for enabling generalization. To test this, we experimented with multiple CoT templates that arranged the core reasoning steps—retrieving candidate lists for each condition, counting the candidates, and stating the final answer—in different orders. The three primary templates evaluated were: (i) **Retrieve-Answer (RA)** retrieves all candidate lists,

Table 4: Final answer and intermediate accuracy on the INTERSECTION and Composition tasks. The results show no OOD generalization across all conditions. For the INTERSECTION task, the order of operations in the CoT program impacts the intermediate trace accuracy on OOD examples.* We tested Count by repeating the entity tokens without relying on value tokens.

| | | | Answer Acc. | | | | Intermediate Acc. | |
| | | | CoT | | non-CoT | | CoT | |
| **Task** | **CoT Program Order** | $k$ | **ID** | **OOD** | **ID** | **OOD** | **ID** | **OOD** |
|---|---|---|---|---|---|---|---|---|
| Intersection | Retrieve > Answer | 2 | 1.00 | 0.01 | 1.00 | 0.00 | 1.00 | 1.00 |
| | | 3 | 1.00 | 0.01 | 1.00 | 0.00 | 1.00 | 1.00 |
| | Count > Retrieve > Answer | 2 | 1.00 | 0.01 | — | — | 1.00 | 0.02 |
| | | 3 | 1.00 | 0.00 | — | — | 1.00 | 0.03 |
| | Retrieve > Count > Answer | 2 | 1.00 | 0.01 | — | — | 1.00 | 0.93 |
| | | 3 | 1.00 | 0.07 | — | — | 1.00 | 0.84 |
| | Retrieve > Count* > Answer | 2 | 1.00 | 0.01 | — | — | 1.00 | 0.93 |
| | | 3 | 1.00 | 0.03 | — | — | 1.00 | 0.84 |
| Composition | Bridge > Compose | 2 | 1.00 | 1.00 | 1.00 | 0.00 | 1.00 | 1.00 |
| | | 3 | 1.00 | 1.00 | 1.00 | 0.00 | 1.00 | 1.00 |

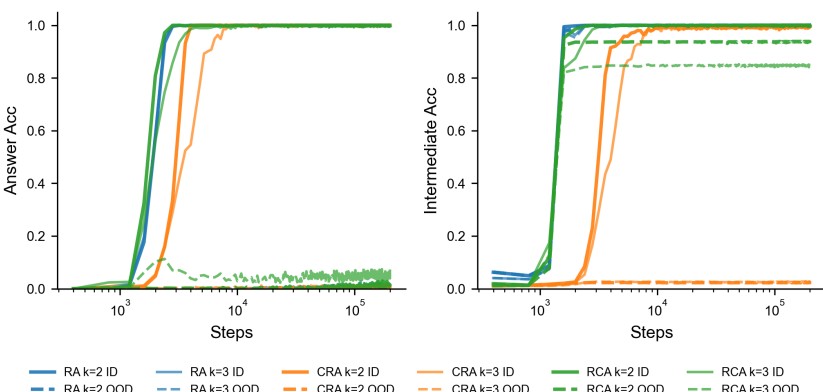

Figure 11: **Learning curves for the INTERSECTION task with $k = 2, 3$.** We compare three CoT templates: Retrieve-Answer (RA), Count-Retrieve-Answer (CRA), and Retrieve-Count-Answer (RCA). **Left Panel (Answer Accuracy):** All models achieve perfect ID accuracy but fail to generalize the final answer to OOD data. **Right Panel (Intermediate Accuracy):** The model's ability to follows the given CoT template varies depending on the location of counting procedure, where the model fails.

then states the final answer; (ii) **Count-Retrieve-Answer (CRA)** states a hint about the count, retrieves lists, then answers; (iii) **Retrieve-Count-Answer (RCA)** retrieves all lists, explicitly counts the intersecting entities, then answers.

The detailed results are presented in Table 4 and visualized in Figure 11. The primary finding is that no CoT template enabled the model to generalize to OOD queries. Across all experimental conditions, the final OOD answer accuracy remained near zero.

## C.5 MORE RESULTS

Figure 12 describes the dynamics of unfaithfulness in a task COMPARISON but with a range of symmetric values. We also show the fit parameters and the goodness of fit in Table 6. Figure 13 shows the learning curves of CoT-guided and non-CoT models on the COMPOSITION task following Wang et al. (2024) setup.

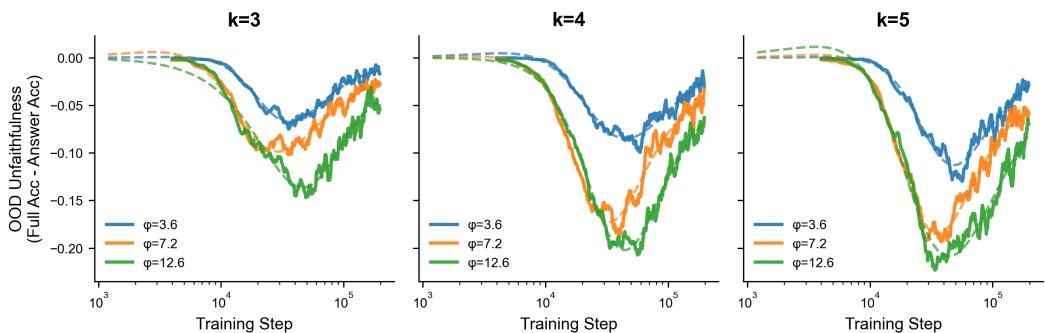

Figure 12: Unfaithfulness gap in the COMPARISON task. Similar to Figure 2 but using value range $v \in [-20, 20]$.

Table 5: Fitted parameters for the COMPARISON task using value range $v \in [0, 20]$, separated by reasoning type (Max vs. Min). The side-by-side layout highlights the primary finding: CoT-guided models consistently achieve a much earlier takeoff point ($t_0$) and higher final accuracy ($L$) than their non-CoT counterparts across all conditions.

| | | | CoT-guided | | | | | non-CoT | | | | |
|---|---|---|---|---|---|---|---|---|---|---|---|---|
| **Ratio ($\phi$)** | $k$ | **Type** | $L$ | $k_{\text{fit}}$ | $t_0$ **(K)** | $R^2$ | **RMSE** | $L$ | $k_{\text{fit}}$ | $t_0$ **(K)** | $R^2$ | **RMSE** |
| 3.6 | 3 | Max | 0.940 | 5.80 | 16 | 0.975 | 0.0352 | 0.597 | 5.14 | 120 | 0.965 | 0.0296 |
| | | Min | 0.937 | 5.68 | 17 | 0.979 | 0.0323 | 0.576 | 5.75 | 118 | 0.971 | 0.0272 |
| | 4 | Max | 0.918 | 4.67 | 20 | 0.964 | 0.0431 | 0.534 | 6.02 | 138 | 0.964 | 0.0258 |
| | | Min | 0.917 | 5.14 | 20 | 0.969 | 0.0406 | 0.406 | 7.22 | 106 | 0.962 | 0.0261 |
| | 5 | Max | 0.921 | 4.67 | 20 | 0.961 | 0.0452 | 0.273 | 7.40 | 137 | 0.923 | 0.0214 |
| | | Min | 0.926 | 4.61 | 20 | 0.959 | 0.0467 | 0.227 | 8.16 | 116 | 0.914 | 0.0223 |
| 7.2 | 3 | Max | 0.964 | 5.99 | 11 | 0.982 | 0.0258 | 0.710 | 5.18 | 90 | 0.974 | 0.0343 |
| | | Min | 0.964 | 5.77 | 11 | 0.980 | 0.0264 | 0.685 | 4.97 | 89 | 0.965 | 0.0379 |
| | 4 | Max | 0.934 | 5.06 | 13 | 0.976 | 0.0306 | 0.585 | 6.05 | 96 | 0.976 | 0.0285 |
| | | Min | 0.936 | 4.99 | 13 | 0.974 | 0.0319 | 0.648 | 5.69 | 111 | 0.971 | 0.0319 |
| | 5 | Max | 0.927 | 5.31 | 15 | 0.979 | 0.0300 | 0.760 | 4.95 | 149 | 0.963 | 0.0326 |
| | | Min | 0.927 | 5.19 | 15 | 0.975 | 0.0330 | 0.931 | 4.76 | 192 | 0.961 | 0.0320 |
| 12.6 | 3 | Max | 0.948 | 5.82 | 12 | 0.981 | 0.0272 | 0.993 | 4.56 | 65 | 0.986 | 0.0343 |
| | | Min | 0.951 | 5.94 | 12 | 0.980 | 0.0278 | 0.907 | 4.95 | 62 | 0.984 | 0.0350 |
| | 4 | Max | 0.914 | 5.87 | 14 | 0.971 | 0.0339 | 1.052 | 4.54 | 86 | 0.987 | 0.0338 |
| | | Min | 0.912 | 5.34 | 14 | 0.975 | 0.0317 | 0.910 | 4.72 | 82 | 0.985 | 0.0321 |
| | 5 | Max | 0.899 | 5.71 | 15 | 0.970 | 0.0356 | 0.984 | 5.19 | 92 | 0.983 | 0.0374 |
| | | Min | 0.905 | 5.08 | 15 | 0.975 | 0.0323 | 0.835 | 5.96 | 75 | 0.988 | 0.0300 |

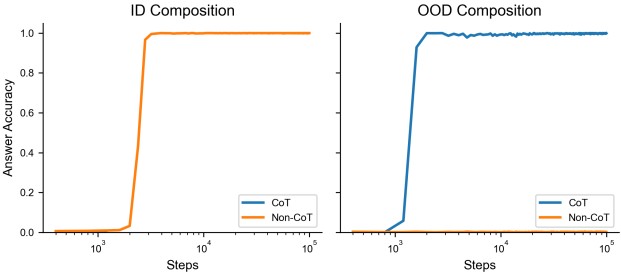

Figure 13: Training curves of the non-CoT and CoT-guided transformer on the COMPOSITION task when $k = 2$. The data generation protocol follows Wang et al. (2024) used in the paper.

Table 6: Similar to Table 5, but with symmetric value range $v \in [-20, 20]$

| Ratio ($\phi$) | $k$ | Type | CoT-guided | | | | | non-CoT | | | | |
|---|---|---|---|---|---|---|---|---|---|---|---|---|
| | | | $L$ | $k_{\text{fit}}$ | $t_0$ (K) | $R^2$ | RMSE | $L$ | $k_{\text{fit}}$ | $t_0$ (K) | $R^2$ | RMSE |
| 3.6 | 3 | Max | 0.947 | 6.38 | 30 | 0.980 | 0.0397 | 0.561 | 5.55 | 147 | 0.958 | 0.0269 |
| | | Min | 0.941 | 5.79 | 29 | 0.978 | 0.0401 | 0.437 | 6.58 | 110 | 0.956 | 0.0281 |
| | 4 | Max | 0.948 | 6.09 | 35 | 0.976 | 0.0447 | 0.458 | 5.59 | 198 | 0.937 | 0.0197 |
| | | Min | 0.933 | 6.00 | 38 | 0.977 | 0.0442 | 0.355 | 6.08 | 175 | 0.918 | 0.0206 |
| | 5 | Max | 0.944 | 6.33 | 36 | 0.975 | 0.0472 | 0.183 | 8.02 | 160 | 0.877 | 0.0157 |
| | | Min | 0.935 | 6.29 | 37 | 0.978 | 0.0439 | 0.151 | 8.12 | 152 | 0.876 | 0.0141 |
| 7.2 | 3 | Max | 0.959 | 5.17 | 24 | 0.976 | 0.0397 | 0.544 | 5.67 | 91 | 0.969 | 0.0294 |
| | | Min | 0.970 | 4.94 | 24 | 0.976 | 0.0399 | 0.464 | 5.33 | 84 | 0.948 | 0.0332 |
| | 4 | Max | 0.961 | 5.24 | 28 | 0.973 | 0.0445 | 0.398 | 5.13 | 126 | 0.940 | 0.0253 |
| | | Min | 0.953 | 4.90 | 28 | 0.969 | 0.0460 | 0.369 | 4.55 | 130 | 0.921 | 0.0253 |
| | 5 | Max | 0.965 | 5.35 | 28 | 0.972 | 0.0455 | 0.382 | 4.98 | 205 | 0.909 | 0.0190 |
| | | Min | 0.952 | 5.05 | 30 | 0.969 | 0.0477 | 0.193 | 7.95 | 114 | 0.921 | 0.0178 |
| 12.6 | 3 | Max | 0.945 | 5.20 | 25 | 0.970 | 0.0446 | 0.829 | 3.85 | 154 | 0.960 | 0.0329 |
| | | Min | 0.951 | 4.79 | 25 | 0.965 | 0.0475 | 0.711 | 3.72 | 114 | 0.961 | 0.0328 |
| | 4 | Max | 0.948 | 4.59 | 30 | 0.962 | 0.0511 | 0.619 | 4.07 | 169 | 0.933 | 0.0311 |
| | | Min | 0.931 | 4.98 | 30 | 0.960 | 0.0523 | 0.386 | 5.32 | 106 | 0.958 | 0.0228 |
| | 5 | Max | 0.936 | 4.89 | 28 | 0.960 | 0.0518 | 0.945 | 4.10 | 280 | 0.950 | 0.0243 |
| | | Min | 0.941 | 4.74 | 32 | 0.954 | 0.0577 | 0.415 | 4.60 | 157 | 0.924 | 0.0245 |

# D  MECHANISTIC PROBING OF INTERNAL REPRESENTATIONS

## D.1  LINEAR PROBING

Beyond end-to-end performance, we seek to understand the internal computational mechanisms by which the model solves these reasoning tasks. To this end, we employ linear probing to analyze the information encoded in the hidden states of the frozen, pre-trained transformer at each layer. Formally, let $\mathbf{h}_t^{(l)} \in \mathbb{R}^{d_{\text{model}}}$ be the hidden state vector at token position $t$ for layer $l \in \{0, \ldots, L-1\}$. A linear probe $P$ with parameters $(W_P, b_P)$ is trained to predict a target label $y$ by minimizing a cross-entropy loss on the output of $P(\mathbf{h}_t^{(l)}) = \text{softmax}(W_P \mathbf{h}_t^{(l)} + b_P)$. We design two distinct probes to trace the flow of information:

**Answer Probe**  assesses if the final answer has been computed. We train it on the hidden state at the final token position of the query, $t_{\text{final}}$. For SORTING task requiring multiple answer tokens, we predict the first answer token. The objective is to predict the correct winning entity that constitutes the answer, $y_{\text{ans}} = \mathcal{E}(f_{\text{task}}(q))$, where $\mathcal{E}$ maps entity tokens to a class index. High accuracy indicates the reasoning is complete at layer $l$.

**Fact Probe**  assesses if the model has retrieved raw factual knowledge. We train it on the hidden state corresponding to the position of an entity token $e_i$ within the query, denoted $t_{e_i}$. The probe's objective is to predict the correct value associated with that entity for the queried attribute, $y_{\text{val}} = \mathcal{V}(\mathcal{KB}_{\text{attr}}(e_i, a_j))$, where $\mathcal{V}$ maps value tokens to a class index. High accuracy indicates successful knowledge retrieval at layer $l$.

## D.2  CAUSAL TRACING

To move beyond identifying informational correlation and instead study the specific representations causally responsible for the model's computations, we employ Causal Tracing via activation patching, allowing us to isolate the contribution of specific token representations at each layer to the final output. First, we run the pre-trained model on a clean input prompt $P_{\text{clean}}$ and record the log-probability of the ground-truth answer, $\log p(y_{\text{ans}}|P_{\text{clean}})$. Second, we run a corrupted input $P_{\text{corrupt}}$,

where a token in the query is replaced. We then run the model on this corrupted input. At last, we perform a forward pass on the clean query, but at a specific layer $l$, we patch the hidden state.

Formally, let $\mathbf{H}^{(l,\text{clean})} = (\mathbf{h}_0^{(l)}, \ldots, \mathbf{h}_T^{(l)})$ be the sequence of hidden states at layer $l$ for the clean run. We create a patched sequence, $\mathbf{H}^{(l,\text{patched})}$, by replacing the hidden state at a single position $t_{\text{patch}}$ with the corresponding hidden state from the corrupted run:

$$\mathbf{h}_t^{(l,\text{patched})} = \begin{cases} \mathbf{h}_t^{(l,\text{corrupt})} & \text{if } t = t_{\text{patch}} \\ \mathbf{h}_t^{(l,\text{clean})} & \text{if } t \neq t_{\text{patch}} \end{cases}$$

This patched sequence $\mathbf{H}^{(l,\text{patched})}$ is then propagated through the remaining layers of the model, from $l$ to $L-1$, to produce a final patched probability distribution over the vocabulary. The causal effect $\mathcal{C}$ of the representation at position $t_{\text{patch}}$ and layer $l$ is defined as the degradation in the log-probability of the correct answer after the intervention: $\mathcal{C}(l, t_{\text{patch}}) = \log p(y_{\text{ans}}|P_{\text{clean}}) - \log p(y_{\text{ans}}|P_{\text{patched}})$. A positive effect indicates that the representation at $(l, t_{\text{patch}})$ was causally necessary to produce the correct answer.

## E LANGUAGE MODEL USAGE

We used language models to help improve grammar and paper writing. We also used the language models to review the draft for feedbacks.

