# OpenReview forum: "The Kinetics of Reasoning: How Chain-of-Thought Shapes Learning in Transformers?"
_ICLR.cc/2026/Conference — Submitted to ICLR 2026_

### Official Review · Reviewer_1smr · 2025-10-31

**Soundness:** 2
**Presentation:** 2
**Contribution:** 2
**Rating:** 2
**Confidence:** 3

**Summary:**

The paper studies how chain-of-thought (CoT) supervision changes the way transformers learn to reason in a controlled, synthetic setup. All experiments are built on a procedurally generated knowledge base (KB) consisting of entities, attributes, and binary relations, which provides the structured symbolic world for reasoning. Specifically, they train 12-layer GPT-2–style transformers from scratch on four symbolic tasks: Comparison (given entities and their attributes, decide which one ranks highest), sorting (order a list of entities by an attribute), intersection (find the entity common to multiple sets), and composition (apply a sequence of relations to infer a new entity). Each of these tasks provides a complexity parameter and an ID/OOD split. Here, ID examples use one set of entity names during training, while OOD examples use held-out entity names but the same structural forms and sequence lengths. In addition, each task has two output formats: (1) direct answer only, and (2) CoT-guided output that includes intermediate steps and the answer.

The authors find that models achieve perfect ID performance with both output formats, but CoT enables OOD generalisation on two tasks (sorting, composition) where non-CoT does not generalise. For intersection, both formats fail to generalise. Next, the authors attempt to quantify learning dynamics by fitting the OOD accuracy curves (in log training steps) with a 3-parameter logistic model. Those parameters are: the ceiling accuracy (the asymptotic OOD performance once learning has converged), the take-off point (the log training step where accuracy reaches half of the ceiling accuracy), and a slope parameters. They find that the extracted parameters behave systematically with task difficulty and the data ratio. They interpret these shifts as evidence that task difficulty increases the effective barrier to generalisation, while more data and CoT supervision make models generalise earlier and reach higher final OOD accuracy.

The authors also track answer and trace accuracy during training and observe what they term the „unfaithfulness gap“: a transient unfaithfulness phase where the answers are correct but the traces incorrect, before traces and answers eventually align. Finally, the authors perform linear probing (predict answer given hidden state) and activation patching to get insights into the computations of the models. They find that the CoT models encode information about the answer much earlier in the model, suggesting that CoT supervision shapes internal computations.

**Strengths:**

- The experiments isolate the effect of chain-of-thought supervision by training identical transformers with and without CoT traces on the same symbolic tasks.
- Fitting OOD accuracy over log training steps with a logistic model an interesting idea to quantify learning speed and saturation. This kinetic framing is novel.

**Weaknesses:**

- The paper’s “OOD” split tests whether models generalise to new entity names under the same reasoning depth, but not to longer or more complex reasoning chains. This setup mainly compares memorisation against symbol abstraction. Since CoT is theoretically expected to help with length generalisation [1], it would have been much more interesting to evaluate generalisation to increased reasoning depth.
- The observed generalisation improvements of CoT over non-CoT may reflect a shift from memorising entity-answer pairs to learning reusable sequence patterns. In other words, CoT might simply force the model to attend to the relational structure of the trace rather than to specific symbols, which naturally enables transfer to unseen entities. It would therefore be interesting to see whether CoT enables algorithmic induction or just lexical abstraction (e.g., by testing extrapolation to unseen reasoning depths).
- The „faithfulness“ analysis appears to define trace correctness as exact match with the ground-truth trace. This would mean that even minor formatting difference (e.g. swapping order, missing comma, or semantically equivalent but syntactically different steps) count as unfaithful. Consequentially, the observed „unfaithfulness phase“ might just reflect formatting inconsistencies rather than truly incorrect reasoning. Including qualitative examples of mismatched traces or alternative metrics would help clarify this.
- In the causal tracing experiments, the observed computational pattern in the CoT model (Fig. 7) is interpreted as evidence that they adopt the „correct computational structure“. However, the tracing plots for CoT Comparison in Fig. 6 are noisy and not clearly smooth, despite achieving near-perfect generalisation. It is therefore unclear why the pattern is evidence for algorithmic correctness and appears like a post-hoc conclusion.

[1] W. Merrill and A. Sabharwal, ‘The Expressive Power of Transformers with Chain of Thought’, in The Twelfth International Conference on Learning Representations, 2024.

**Questions:**

- Have you evaluated whether CoT also improves length or compositional generalisation?
- In the faithfulness analysis, how often are traces “unfaithful” because of minor formatting errors versus semantically wrong reasoning? Could you provide qualitative examples of what constitutes an incorrect trace?
- Can your kinetic “laws” be used predictively? That is, given task complexity or data ratio, can you forecast the approximate take-off step t_0 or ceiling L? Or are these relationships purely descriptive? If predictive, showing out-of-sample fits or held-out task extrapolations would make this argument much stronger.

---

> ### Author Response · Authors · 2025-11-21
> **Reply to Reviewer 1smr (1/2)**
>
> We thank Reviewer 1smr for their thorough review and for highlighting the core strengths of our paper, namely the **novelty of our "kinetic framing"** for logistic modeling and our **controlled experimental design** that "isolate[s] the effect of chain-of-thought supervision." We appreciate the insightful questions and address them below.
>
> **W1/Q1: Symbol Abstraction vs. Length Generalization**: \
> The reviewer is correct that our ‘OOD‘ split is designed to test the generalization of language models by **preventing memorization**. We aim to test **symbol abstraction** instead of ‘length’ generalization, which means the capability of learning a reasoning pattern of a fixed complexity ($k$) and applying this learned pattern to unseen entities. We chose this setup for two reasons: (1) As our results demonstrate, symbol abstraction is highly non-trivial and reflective of learning dynamics/kinetics. Direct-answer models can fail on this task completely (e.g. 18% acc. on Sorting and 0% on Composition), despite the models having remembered all atomic facts. This difficulty itself has already demonstrated that a model must first master symbolic abstraction before generalizing to longer chains. Also, we chose to build upon experimental protocols of prior mechanistic studies [1,2] to make sure our results are comparable to the field.
>
> We kindly ask the Reviewer 1smr to clarify and confirm *‘Since CoT is theoretically expected to help with length generalisation [3]’* claim. To our best knowledge, Merrill et al. studied the relationship between Transformer expressivity and output sequence length, and the authors did not discuss ‘length generalisation’.
>
> **W2: Mechanism of Improvement** \
> The reviewer's description is a good articulation of our core findings. Our experimental design, which trains on ID entities and tests on OOD entities, is specifically designed to differentiate what the reviewer terms "memorising entity-answer pairs" from "learning reusable sequence patterns". The **non-CoT model** does fail by simply memorizing, which is most evident in the Composition task, where it achieves perfect ID accuracy but near-zero OOD accuracy. It learns the entity-answer pairs but fails to abstract the underlying reasoning pattern. The **CoT-guided model** successfully learns the "reusable sequence pattern" to achieve high OOD generalization (92% on Sorting, 100% on Composition, etc.) We would like to clarify that the CoT-guided model would also need to generate the trace given the same input (query) to both direct-answer and CoT-guided model.
>
> **W3/Q2: Validity of Strict Trace Matching** \
> We would like to clarify that we symbolized the reasoning tasks in terms of entities, attributes, and values. Such symbols are represented as separate tokens and there is no ‘formatting difference’, as we repeatedly explained in Sec. 3, Table 1, and Appendix A. The generation of reasoning problems follows fixed templates, where the exact match is defined based on different tasks, e.g. sorting requires the entities and corresponding values are sorted in-order. It's worth emphasizing that the order of entity-value pairs is fixed in the training examples, thus the order of intermediate steps (values) corresponds to the order of entities in the prefix. Consider a Comparison query max(age, Alice, Bob) where the atomic facts are (Alice, age, 10) and (Bob, age, 20). The algorithmically defined trace is (10, 20) corresponding to the input entities in-order. If the model generates the (20,10) trace, it implies the model believes (Alice, age, 20) and (Bob, age, 10). For a Sorting query, its trace is defined as the sorted list of values. For a set of distinct values, the sorted order is mathematically unique. Any other order is, by definition, incorrect.
>
> **W4: Interpretation of Causal Traces**\
> We respectfully clarify that the comparison between Fig. 6 and Fig. 7 is not an indication of a post-hoc conclusion, but rather an evidence of a shift in computational mechanism. In tasks that require internal serial computation, such as the Direct-Answer Comparison model (Fig. 6a, left) or the inherent sequential CoT Composition model (Fig. 7), we indeed see a consistent clean ‘serial’ computation, indicating a layer-by-layer computation as we highlighted in Sec 7.1 and 7.2. For CoT-guided Comparison where the reasoning is externalized into the context window through ‘value’ tokens, the ‘noisy’ pattern in Fig. 6a (right) reflects this non-serial mechanism where the model gets the answer from the context rather than computing it serially. This observation echoes our linear probing results (Fig. 5), where the answer becomes decodable from very early layers.

---

> > ### Comment · Reviewer_1smr · 2025-11-27
> >
> > Thank your for your response. I address each point below.
> >
> > W1/2: Thanks for clarifying the scope of your work.
> >
> > Reg. [3]: Merrill et al. build on prior work that shows that "transformer decoders without any intermediate steps can only solve problems that lie inside the fairly small circuit complexity class TC0". They then show that transformers with "[l]inear intermediate steps [can] simulate automata (NC1-complete), which cannot be done without intermediate steps". Importantly, circuit complexity classes are defined over inputs of unbounded length. This implies that, in theory, transformers augmented with chain-of-thought (i.e., linear intermediate steps) can generalise to arbitrarily long inputs for problems in NC1, whereas without such intermediate computation they remain fundamentally limited to TC0. Some of the tasks you study admit parallel algorithms of polylogarithmic depth (e.g., via trees of comparators or sorting networks), placing the corresponding decision problems in NC. Thus, in addition to symbol abstraction, I believe it would have been interesting to study whether CoT helps with length generalisation.
> >
> > W3: Thank you for the clarification. I understand that traces are purely symbolic and thus exact match should reflect semantic rather than formatting differences. I would still find it helpful to see a few qualitative examples of mismatched traces, to better understand the nature of the observed unfaithfulness phase.
> >
> > Q3: Thanks for clarifying. I agree that the logistic model is an interesting contribution and would be excited about future work on this.
> >
> > Based on this, I increase my score to 4.

---

> > > ### Author Response · Authors · 2025-11-28
> > >
> > > We thank the reviewer 1smr for finding our contributions interesting and we are also excited about potential future works built upon our contributions.
> > >
> > > We appreciate the reviewer for sharing their thoughts on the expressivity class and generalization. We also agree that exploring the length generalization in orthogonal to the entity generalization (this work) could be an interesting work in the future.
> > >
> > > We are definitely happy to clarify what would be the *trace mismatch*. Using the Comparison example in Table 1, we are figuring out the max(< height >) of entities (< Alice >< Bob> < Chloe >), with values of < 65 >< 72 >< 68 >. We expect a trace with an answer exactly as: < 65 >< 72 >< 68 >< Bob >, any entity-value mismatch would be considered as incorrect. During the unfaithful phase in the training (Figure 4), most errors are due to wrong retrieved values (e.g., < 65 >< 73 >< 68 >).
> > >
> > > We are dedicated to continuing clarifying any remaining concerns.
> > >
> > > Happy Holiday, \
> > > Authors

---

> ### Author Response · Authors · 2025-11-21
> **Reply to Reviewer 1smr (2/2)**
>
> **Q3: Descriptive vs. Predictive Modeling** \
> We thank the reviewer for sharing this possible direction. This is a critical question regarding the scope of our intended contribution. In its current form, our kinetic model and the arrhenius perspective are descriptive. Our primary goal is to move the understanding of CoT from qualitative statements to a quantitative one. We are the first to demonstrate that the learning dynamics of (CoT-guided) transformers can be accurately described ($R^2 > 0.9$) by a compact logistic model with just three parameters ($L, t_0, k_{fit}$).
> In combination with earlier exploration in neural network grokking literature [4,5], we believe establishing this descriptive kinetic model is a significant contribution in itself, as it provides a necessary taxonomy to quantify how the data complexity ($k$) and data distribution ($\phi$) systematically shape the learning dynamics. We fully agree that establishing a predictive framework is an exciting next step. We will explicitly state in the conclusion that our descriptive framework provides the basis for such future studies.
>
> [1] Wang et al. Grokked Transformers are Implicit Reasoners: A Mechanistic Journey to the Edge of Generalization. NeurIPS 2024 \
> [2] Abramov et al. Grokking in the Wild: Data Augmentation for Real-World Multi-Hop Reasoning with Transformers. ICML 2025 \
> [3] Merrill et al. The Expressive Power of Transformers with Chain of Thought. ICLR 2024. \
> [4] Power et al. Grokking: Generalization Beyond Overfitting on Small Algorithmic Datasets. Arxiv, 2022. \
> [5] Liu et al. Towards Understanding Grokking: An Effective Theory of Representation Learning. NeurIPS 2022.

---

### Official Review · Reviewer_x9ue · 2025-11-01

**Soundness:** 2
**Presentation:** 2
**Contribution:** 2
**Rating:** 4
**Confidence:** 4

**Summary:**

This paper investigates how transformers learn under chain-of-thought (CoT) supervision by analyzing learning dynamics through the lens of grokking. The authors pretrain transformers on a suite of symbolic reasoning tasks (COMPARISON, SORTING, COMPOSITION, INTERSECTION) and compare CoT-guided training (producing intermediate reasoning traces) to direct-answer training. They propose a kinetic framework that models out-of-distribution (OOD) accuracy as a function of log training steps using a three-parameter logistic curve, capturing learning speed and saturation. The analysis reveals that CoT can accelerate generalization, especially for tasks of moderate algorithmic complexity, though both CoT and non-CoT models eventually fail on more complex reasoning tasks. The paper also identifies a transient phase of “trace unfaithfulness,” where models produce correct answers but inconsistent intermediate traces, and uses probing analyses to relate CoT to shifts in internal computation.

**Strengths:**

•	The idea of fitting OOD accuracy over log training steps with a three-parameter logistic model to quantify learning speed and saturation is novel and interesting.
This kinetic framing offers a new lens to analyze how CoT affects learning dynamics and the grokking transition.

**Weaknesses:**

•	Intermediate accuracy metric definition: I understand the intermediate accuracy to be defined as a “trace overlap” measure matching the ground-truth trace to the predicted trace. However, many tasks can be solved using different but valid traces. This metric disregards all expedient but non-identical traces, potentially overstating the “knowledge gap” or “unfaithfulness phase.” A more nuanced metric accounting for alternative valid reasoning paths would strengthen the claims.
•	Limited evaluation scope: All experiments use a single transformer architecture (12-layer GPT-2 style) and a single random seed. This narrow evaluation is insufficient to support the general claims about CoT-induced learning dynamics and kinetics. Even if trends appear robust, more empirical evidence (e.g., across seeds and architectures) is needed.
•	Inconsistent measurement axes: It is unclear why training steps are used for some comparisons while FLOPs are used for others (e.g., Figure 3). The authors should clarify whether these metrics are interchangeable or whether differences in scaling affect the observed kinetic fits. The paper reports a significant drop in OOD performance between the COMPOSITION and SORTING tasks despite similar accuracy-over-log-step curves. The connection between the kinetic fit and this performance difference is not clearly explained.
•	Readability and structure: The paper is dense and often difficult to follow. The narrative flow between theoretical framing, kinetic modeling, and empirical evidence is not always clear, making it hard to track the central argument.

**Questions:**

1.	Does the “intermediate accuracy” metric consider different syntactic but semantically equivalent reasoning traces? If not, how might this affect the interpretation of unfaithfulness?
2.	In Figure 3, are the indices of the ϕ labels identical to the task parameters listed in Table 3?
3.	Can you clarify the meaning and practical interpretation of the data ratio (ϕ)?
4.	Why are training steps used for comparison in some plots but FLOPs in others (e.g., Figure 9)? Are the fitted curves consistent across both scales?
5.	Could the “knowledge gap” phenomenon be partially explained by the existence of multiple correct traces that differ from the ground-truth trace?

---

> ### Author Response · Authors · 2025-11-21
> **Reply to Reviewer x9ue (1/2)**
>
> We thank reviewer x9ue for the constructive and insightful feedback. We are encouraged that the reviewer acknowledged our contributions including **‘kinetic framing’** and **‘three-parameter logistic model…’** are **interesting and novel**.  We appreciate the reviewer's assessment and are confident that we can fully address these concerns below.
>
> **W1/Q1/Q5: Validity of Strict Trace Matching** \
> We thank the reviewer for this concern.  The reviewer is correct that our intermediate (tracy) accuracy is a **strict in-order match** with the ground-truth trace. In our controlled symbolic setup, there are no *‘different syntactic but semantically equivalent’* reasoning traces (e.g., formatting difference). Unlike open-ended natural language reasoning, our tasks are symbolic (Sec 3, Table 1) and unambiguous (Sec 3.2, Appendix A). Any deviation from the ground truth is a solid error. Consider a Comparison query max(age, Alice, Bob) where the atomic facts are (Alice, age, 10) and (Bob, age, 20). The algorithmically defined trace is (10, 20) corresponding to the input entities in-order. If the model generates the (20,10) trace, it implies the model believes (Alice, age, 20) and (Bob, age, 10). This is a factual hallucination instead of a formatting error. For a Sorting query, its trace is defined as the sorted list of values. For a set of distinct values, the sorted order is mathematically unique. Any other order is, by definition, incorrect.
>
> Therefore, the ‘unfaithfulness gap’ we observe is not an artifact of evaluation metrics. This gap represents a real phenomenon where the model predicts the final answer correctly via shortcuts, but fails to generate the correct intermediate steps (traces) to compute the answer. We will clarify in Sec 3.3 and Table 1.
>
> **W2: Scope of Evaluation** \
> We thank the reviewer for this valid point regarding the evaluation scope. We would like to clarify our two lines of reasoning on this: \
> (1) We respectfully point to our architectural control experiment in Figure 10 and Appendix C.3, as we articulated Sec 4 (line 256-259). We *did* test an alternative architecture (Mamba) with a matched parameter count. The finding that **Mamba failed to generalize** on the Comparison task (even with CoT guidance) while the Transformer succeeded is an important result, suggesting the observed learning dynamics are not universal to all architectures. \
> (2) This is a fair point, and we will explicitly add it as a limitation. However, we are confident in our findings because the observed trends are **highly robust and systematic across different tasks** (Comparison, Sorting) and multiple complexities ($k=3, 4, 5$).
>
> **W3: Consistency of Measurement (Steps vs. FLOPs)** \
> This is not an inconsistency but an **necessary choice** to answer two different questions. The axes are **not interchangeable**. As mentioned in Sec 3.3 (line 190-194), we trained the same model architecture using ‘Direct-Answer’ and ‘CoT-guided’ formats. For each token, the cost is the same. **Training Steps** is our primary axis for analyzing **learning dynamics**. Our kinetic model, including $t_0$, is defined in terms of training steps. This is the "time" variable for our kinetic/dynamics analysis. FLOPs is used specifically in Fig. 3 (right) to make a separate point about **compute efficiency**. As a CoT-guided model generates more tokens, we intentionally added this FLOPs comparison to compare ‘Direct-Answer’ and ‘CoT-guided’ under the same training budget.
>
> The model performance on the Composition and Sorting tasks are not directly comparable. Though, the similar S-shape curves indicate the generalizability of the kinetic model explanation across different reasoning tasks. If the reviewer refers to performance differences between non-CoT and CoT models, we indicated the belongings of training curves in the legend of Fig. 3. We thank the reviewer for raising this concern and we will clarify it in our paper.

---

> > ### Author Response · Authors · 2025-11-24
> > **Reply to Reviewer x9ue (2/2)**
> >
> > **W4: Writing Style**\
> > Hope our clarifications above have addressed your concerns. We are happy to answer any further questions you may have to help you understand our paper better.
> >
> > **Q2/Q3: Definition of Data Ratio** \
> > Yes, the $\phi$ labels (data ratios = 3.6, 7.2, 12.6) are consistent across all experiments. The difference in presentation is simply the task being analyzed: Fig. 3 (left) shows the Comparison task, while Table 3 analyzes the Sorting task.  As defined in Eq.1 (Sec 3), $\phi$ is the data ratio of composed reasoning examples to all atomic facts that composed the underlying database. It’s the ratio of composed reasoning examples (the training queries, $D_{train}$) to the atomic facts in the knowledge base ($F_{base}$). Since we keep the size of the knowledge base fixed across varying setups ($k, \phi$), phi controls the data distribution. A higher $\phi$ means the model sees more unique reasoning examples (combinations) derived from the same set of facts. For instance, if we have a knowledge base with 1000 atomic facts (single-hop) and we construct 3600 multi-hop questions derived from these atomic facts, then $\phi=3.6$.

---

### Official Review · Reviewer_72tx · 2025-11-02

**Soundness:** 2
**Presentation:** 2
**Contribution:** 2
**Rating:** 4
**Confidence:** 2

**Summary:**

This paper studies small transformers on controlled symbolic tasks and claims that CoT supervision accelerates “grokking,” fits OOD accuracy with a 3‑parameter logistic curve in log‑steps, and interprets second‑order trends via an Arrhenius‑style “barrier/temperature” analogy. Empirically, CoT helps on COMPARISON and SORTING but not INTERSECTION; “unfaithfulness” (answers correct while traces are wrong) appears mid‑training; a Mamba control fails to OOD‑generalize.

**Strengths:**

1. Clear, controllable testbed \& measurements. The tasks vary algorithmic difficulty $k$ and data ratio $\phi$; the paper cleanly reports zero-/few-shot OOD accuracy and shows CoT often improves sample-efficiency (incl. a FLOPs view).
2. Architecture control. A matched-size Mamba fails to OOD-generalize while transformers succeed under CoT, highlighting inductive-bias differences.

**Weaknesses:**

1. Limited novel understandings. That CoT accelerates learning and improves expressiveness, and that intersection-like composition is hard, all broadly echo prior CoT understanding. The main new angle is the curve-fitting/Arrhenius narrative, which remains largely phenomenological. The Arrhenius analogy is not stress-tested (no explicit estimation of $\Delta$ or $T_{\text {eff }}$; validation reduces to trending $\hat{r}$ via Eq. (6)).
2. Kinetic self-consistency is shaky. The paper fits a logistic in log-time (Eq. 4) yet explains dynamics via a linear-time logistic ODE and then maps the fitted slope back to a "rate" (Eq. 6). This mixes time scales and treats a log-time shape parameter as a linear-time rate without a principled derivation.
3. "OOD" is closer to compositional recombination than true OOD. Training always includes all atomic facts (ID and OOD) and holds out only composed $k$-hop examples for OOD; test is "compositions of known atoms," not unseen entities/attributes. This inflates how far the conclusions generalize.
4. Evaluation choices may blur compute/efficiency claims. FLOPs are proxied by token counts ("relative FLOPs"), and plots lack error bars/seed variation; some tasks use partial-credit answer scoring while "full-sequence" requires exact trace+answer, complicating cross-task comparisons and the magnitude of the "unfaithfulness" gap.

**Questions:**

See weaknesses

---

> ### Author Response · Authors · 2025-11-21
> **Reply to Reviewer 72tx (1/2)**
>
> We thank the reviewer 72tx for detailed technical feedback. We are encouraged that the reviewer recognized the strengths of our paper in particular **‘clear, controllable testbeds and measurements’**. We appreciate reviewer’s assessment and are confident that we can fully address these concerns as below.
>
> **W1: Regarding Phenomenological Modeling**\
> We thank the reviewer for this critique and we acknowledge that our model is descriptive. But we respectfully argue that establishing a rigorous phenomenological description is a critical and currently missing step in the study of language model learning dynamics. Though the reviewer views our kinetic model as ‘curve-fitting’, we found the description of this **three-parameter** logistic model to the transformer learning dynamics very well. Such a simple model **robustly** fits the complex learning dynamics of transformers with high precision ($R^2 > 0.9$, Table 3) in **many different setups**. Its robustness and generalizability across different tasks complexities itself is a significant discovery. It proves that (CoT-induced) transformer grokking is not random, but follows predictable dynamics that can be **statistically described**.
>
> We agree that the Arrhenius perspective is an *interpretive framework* rather than a fitted regression model. We did not aim to derive numerical constants for "barrier" ($\Delta$) or "temperature" ($T_{eff}$). Instead, Eq. 5 serves to **link** the fitted kinetic parameters (from Eq. 4) to the **experimental controls** ($k, \phi$). The rigor of this perspective lies in the systematic trend reported in Table 3. For example, as the data complexity ($k$) increases, the fitted $t_0$ delays significantly, similar to an increasing energy barrier. As the data ratio ($\phi$) increases, $t_0$ reduces, similar to increased temperature. We will clarify in Sec 5 that Eq.4 is the quantitative tool, while Eq.5 is the conceptual framework for understanding.
>
> **W2: Mathematical Consistency (Log-time vs. Linear ODE)** \
> We thank the reviewer for mentioning the relationship between linear-time ODE (Eq.3) and the log-time observations (Eq.4). We respectfully remind the reviewer that analyzing learning dynamics in log-time is the standard convention in the field instead of an arbitrary choice. In earlier literatures including neural scaling laws [1,2] and grokking series [3,4,5,6,...], the observations are better described in log-scale for network training, consistently demonstrate that learning dynamics and phase transitions occur over orders of magnitude. As we state in the paper, our empirical learning curves *‘fit more parsimoniously’* in log-time. Modeling the learning dynamics and kinetics in linear time would fail to capture the **relevant timescale** of the phenomenon.
>
> The relationship between the linear-time ODE (Eq.3) and the log-time observations (Eq.4) can be easily derived using a chain rule:
> 1. let $x = \log_{10}(t)$ be the log-time timsteps. The derivative of time with respect to the log-scale is $\frac{dt}{dx} = t \ln(10)$.
> 2. The log-time rate of change is therefore: $\frac{dAcc}{dx} = \frac{dAcc}{dt} \cdot \frac{dt}{dx} = r_{\text{linear}} \cdot (t \ln 10)$
> 3. This shows that the log-time slope ($k_{fit}$) is proportional to the linear rate scaled by time.
> 4. By rearranging the terms, we recover the intrinsic linear rate: $\hat{r} \propto \frac{k_{fit}}{t_0 \ln 10}$
>
> We will clarify this chain rule in Sec 5 and App B to ensure the self-consistency is clear.
>
> **W3: Clarifying the Definition of OOD** \
> We thank the reviewer for this precise summary of our setup. We confirm that our ‘OOD’ setting is indeed the compositional generalization where the model needs to apply a learned reasoning pattern to new compositions of known atomic facts, rather than ‘zero-shot knowledge acquisition’ (entirely unknown entities).
>
> To test if a model has learned the underlying reasoning pattern (Comparison/Sorting/etc.), we must ensure the model failures are not contributed from the missing knowledge. If the model has never seen the atomic facts (‘Mary, age, 18’, ‘Alice, age, 19’), it is impossible to compare (‘Who is younger? Mary or Alice?’). By providing all atomic facts, we ensure that any failure on the OOD set is purely a failure of learning the underlying reasoning pattern.
>
> This split/definition of OOD is a standard convention in earlier literatures [5,6]. We are building upon established protocols to ensure our results are rigorous and comparable to the field.

---

> ### Author Response · Authors · 2025-11-21
> **Reply to Reviewer 72tx (2/2)**
>
> **W4: Evaluation Protocols (FLOPs & Partial Credit)** \
> We thank the reviewer for examining our evaluation protocol. We believe our choices are robust and grounded to meet our goals of analysis.
>
> The reviewer is correct that we use token counts as a proxy for FLOPs, where we use FLOPs = 6ND (N is the number of model parameters, and D is the number of tokens generated, [7]). We would like to clarify that, for a fixed model architecture (as we stated in Sec 3.3, line 190-192), N is constant. We use this in Fig. 3 (right) specifically to verify the efficiency gain of CoT is real and not an artifact of generating more tokens.
>
> The reviewer raises a concern that using ‘partial’ correctness versus ‘exact’ correctness might inflate the unfaithfulness gap. We argue this design is necessary to detect unfaithfulness. In our definition, ‘unfaithfulness’ is defined as the model ‘getting the answer right (or mostly right) for the wrong reasons.
>
> For Comparison, the answer is a **single unique entity**, so answer accuracy is naturally binary (exact match). For Sorting, requiring an exact-match on the ordered entity set (more than one entity) would result in near-zero accuracy for the initial phases of the training, masking the underlying learning dynamics completely. By allowing partial credit for the Answer, we can detect when the model has learned the output pattern but failed in the intermediate steps, where the model approximates the output statistically without mastering the underlying reasoning pattern.
>
> We acknowledge the limitation of using a single seed for the consideration of the cost, and we will state this in the final version. However, we are confident that our discoveries are not due to statistical noise for the consistent and robust trends across all the tasks and setups. The same three-parameter logistic kinetic model (Eq.4) accurately fits ($R^2>0.9$) the dynamics across Comparison and Sorting, with training steps and FLOPs, varying data complexities ($k = 3, 4, 5$) , and varying data distributions ($\phi = 3.6, 7.2, 12.6$). The systematic variation of parameters across all these conditions demonstrate that we are observing fundamental rules in the learning dynamics instead of statistical noises.
>
> [1] Kaplan et al. Scaling Laws for Neural Language Models. Arxiv, 2020. \
> [2] Hoffmann et al. Training Compute-Optimal Large Language Models. Arxiv, 2022. \
> [3] Power et al. Grokking: Generalization Beyond Overfitting on Small Algorithmic Datasets. Arxiv, 2022. \
> [4] Liu et al. Towards Understanding Grokking: An Effective Theory of Representation Learning. NeurIPS 2022. \
> [5] Wang et al. Grokked Transformers are Implicit Reasoners: A Mechanistic Journey to the Edge of Generalization. NeurIPS 2024. \
> [6] Abramov et al. Grokking in the Wild: Data Augmentation for Real-World Multi-Hop Reasoning with Transformers. ICML 2025. \
> [7] Austin et al., "How to Scale Your Model", Google DeepMind, online, 2025.

---

### Official Review · Reviewer_JwyN · 2025-11-03

**Soundness:** 3
**Presentation:** 3
**Contribution:** 2
**Rating:** 2
**Confidence:** 4

**Summary:**

The paper studies learning dynamics and generalization of algorithmic tasks with LLMs that use either direct prediction or CoT on four synthetic tasks (intersection, sorting, comparison, composition). It is found that CoT generalizes better, has a phase transition, and reasoning unfaithfulness is reduced as training progresses.

**Strengths:**

Writing:
- The paper is written clearly and is easy to follow.

Conceptual:
- The paper (re-)discovers a few interesting facts, namely reasoning unfaithfulness, CoT helping in generalization and a grokking transition.
- The experiments cover a number of challenging (although synthetic) reasoning tasks and are performed correctly.
- The interpretations of the training curves are interesting, seem plausible and give insight into the training behavior.

**Weaknesses:**

Contribution:
- The paper basically does a number of experiments and reports training curves on them.
- The take home message is, I would assume, well known lore in the deep learning community, including grokking, CoT faithfulness and CoT generalizability. No new insight has been produced.
- The overall setup is highly artificial with only synthetic tasks trained on an LLM from scratch. I am not sure that these findings would translate to pre-trained LLMs that might already exhibit different behaviors.
- The learning curves that are fitted are nice, but again I do not see the significance in them for a wider audience. In order for them to play a similar role to scaling laws in LLM training one would have to train on a wider range of realistic training tasks on pre-trained LLMs. As they are, they are just post-hoc fitting of training dynamics.
- The mechanistic interpretability results feel like a random addition to the main paper topic and consist of standard application of linear probing and causal tracing.

**Questions:**

- What is the significance of your training findings outside the synthetic reasoning tasks for real-world pre-trained LLMs and real reasoning tasks?

---

> ### Author Response · Authors · 2025-11-21
> **Reply to Reviewer JwyN**
>
> We thank Reviewer JwyN for their insightful and positive feedback on our paper's writing, presentation, and the **"interesting"** and **"plausible"** insights into the model training process. We also appreciate the opportunity to clarify contributions and methodological choices of our works.
>
> **W1/W2: Regarding Novelty and Methodology** \
> We would respectfully disagree with the reviewer's comments to our work: "basically... experiments and reporting training curves." Our work follows in the methodological spirit of earlier studies, such as those on neural scaling laws [1,2], which derive fundamental principles from careful analysis of training curves and their fittings. The *"post-hoc fitting"* is in fact our primary contribution: we are the first to propose a quantitative explainable model for the grokking phase transition in both non-CoT and CoT-guided setups.
>
> The reviewer suggests our take-home messages are "well known lore." While the *qualitative* idea that "CoT helps" is known, our paper's novel insights are in the **quantitative mechanism**, and a direct evidence of how the data distribution/complexity relates to the language model learning. To the best of our knowledge, no prior work has provided a clear demonstration of the learning, grokking, and faithfulness dynamics of CoT-augmented models **trained from scratch** in such a clean and testable reasoning setup.
>
> **W3: Justification for Synthetic Data**\
> We appreciate the reviewer's concern about the *"highly artificial"* setup. This was a **necessary experimental design** choice for three reasons: (1) **Studying Learning**: Our paper is about the **learning dynamics and kinetics** One cannot study these fundamental learning dynamics on pre-trained models, since trained models have *already* trained in the first place, and we cannot control their training materials. More importantly, this introduces extra compounding effects. (2) **Experimental Control**: Using synthetic data is the only way to *guarantee* no data leakage and to strictly test out-of-distribution generalization. (3) **Established Methodology**: This controlled from-scratch approach is standard in mechanistic and grokking literature (e.g., [3, 4, 5]), as it is the most effective way to isolate computational mechanisms.
>
> **W4: Purpose of Mechanistic Interpretability** \
> We respectfully clarify that the mechanistic results are not an *"addition"* but an **important component** of our argument. Results described in Sec 4,5 and 6 are outcome (correctness)-oriented, where the probing and tracing (Sec 7) help explain **whether** the internal computations change with the CoT guidance. This follows the standard practice of using probing to explain model behavior, as seen in a prior study [5].
>
> [1] Kaplan et al. Scaling Laws for Neural Language Models. Arxiv, 2020. \
> [2] Hoffmann et al. Training Compute-Optimal Large Language Models. Arxiv, 2022. \
> [3] Power et al. Grokking: Generalization Beyond Overfitting on Small Algorithmic Datasets. Arxiv, 2022. \
> [4] Liu et al. Towards Understanding Grokking: An Effective Theory of Representation Learning. NeurIPS 2022. \
> [5] Wang et al. Grokked Transformers are Implicit Reasoners: A Mechanistic Journey to the Edge of Generalization. NeurIPS 2024.

---

### Author Response · Authors · 2025-11-24
**Global Response**

We thank all reviewers for their time and insightful comments. We are encouraged that reviewers found our writing clear and easy to follow, our experimental setup rigorous, and our learning dynamics/kinetics perspectives novel. We summarize our response to three common questions raised across the reviews.

**1.Novelty and Contribution**\
A common concern (R-JwyN/72tx) was that ideas like ‘CoT help generalize’ are well-known. We respectfully clarify that our contributions are not these **qualitative** heuristics, but providing a set of **quantitative** observations that demonstrate how (reasoning) language models learn to reason in general using a verifiable framework. We found our discoveries novel and robust, as the learning dynamics/kinetics across all our proposed tasks and difficulties can be accurately described ($R^2>0.9$) using a logistic model with just three parameters. And these fitted parameters shift systematically with respect to data distribution/complexity.

**2.Necessity of Synthetic Data with Control**\
Reviewers (R-JwyN/72tx/1smr) questioned the significance of synthetic data and OOD definition. We emphasize that our goal is to study the **learning dynamics/kinetics** from first principles and we would like to isolate compounding effects.

Pre-trained models have unknown training data, making it hard to separate ‘learning a new reasoning pattern’ from ‘memorizing data shortcuts’. Our setup aligns with established methodologies in mechanistic interpretability and grokking researches [1,2,3], where synthetic tasks are the standard for isolating whether a model can learn the underlying reasoning pattern.

**3.Unfaithfulness and Trace Evaluation**\
Reviewers (R-72tx/x9ue/1smr) raised questions regarding our evaluation metrics. Specifically, whether *in-order* strict trace match penalize *’semantically equivalent but syntactically different’* traces (R-x9ue/1smr) and whether partial answer accuracy inflates the unfaithfulness (R-72tx).

We respectfully remind the reviewers that our tasks are symbolic and algorithmic (e.g. Sorting). And the ‘intermediate trace’ is well-defined (e.g. a sorted list of unique values). Thus, a mis-ordered trace is a true error instead of *‘formatting difference’* (R-x9ue/1smr). We also would like to respectfully clarify *‘unfaithfulness inflation’* (R-72tx). Applying strict answer match (in Sorting) would ignore the earlier learning phase where the model gets the answer mostly right for wrong reasons. And we also would like to clarify that we would like to study **how a language model statistically predicts answer first, and then aligns the intermediate steps** this phenomenon itself,  which is already evident and self-contained in Comparison tasks (answer is a unique entity).

We hope these responses addressed the core concerns and we have replied specifically to each reviewer. And we are also happy to engage in further discussions.

[1] Power et al. Grokking: Generalization Beyond Overfitting on Small Algorithmic Datasets. Arxiv, 2022. \
[2] Liu et al. Towards Understanding Grokking: An Effective Theory of Representation Learning. NeurIPS 2022. \
[3] Wang et al. Grokked Transformers are Implicit Reasoners: A Mechanistic Journey to the Edge of Generalization. NeurIPS 2024.

---

### Meta-Review · Area_Chair_mTpK · 2025-12-18

**Summary:**

This paper studies how CoT shapes learning in transformers through the lens of grokking. In particular, the authors perform a series of controlled experiments on symbolic reasoning tasks (comparison, sorting, intersection, composition) and compare CoT with training using direct answers. The results demonstrate that CoT helps for easier tasks (comparison, sorting, composition), but it is still insufficient for more complex ones (intersection). A transient state of "trace unfaithfulness" is also identified, before the model aligns with CoT reasoning.

Overall, the reviewers appreciate the setup and the interpretations of the results. However, they also noted the limited novelty in the phenomena described in the paper. While the authors argued that the point of their work is in the quantitative description of such phenomena, the scope of the evaluation remains rather limited and purely descriptive. Thus, I do not foresee the consensus on this paper to become positive (even in the presence of a regular discussion period), and unfortunately I recommend a rejection at this stage.

Having said that, I do think the ideas exposed here are valuable and I encourage the authors to incorporate the reviewer comments in the next iteration to be submitted to a future venue.

**Reviewer Concerns:**

Some of the concerns of the reviewers were addressed in the rebuttal:

* W2 and W4 of Reviewer 72tx.
* W1, Q1, Q2, Q3, Q5, and W3 of Reviewer x9ue.
* Comment on using only a single transformer architecture by Reviewer x9ue.
* W3, Q2 of Reviewer 1smr.

However, other comments are difficult to address without additional experiments and a substantial revision:

* Limited novel understanding, as raised by Reviewers JwyN and 72tx
* Artificial setup with limited scope, as raised by Reviewers JwyN and x9ue.
* Purely descriptive results, as raised by Reviewers 72tx and (to an extent) 1smr.
* Comment on length generalization by Reviewer 1smr.

**Reviewer Scores:**

The score of Reviewer 1smr had already changed from 2 to 4. It is possible that other scores may have been raised but, given the nature of the comments and the corresponding rebuttal, I find it unlikely that the overall consensus on the paper would have shifted towards a recommendation to accept it in its current form.

---

### Decision · Program_Chairs · 2026-01-26

Reject